# WIKI-R1: INCENTIVIZING MULTIMODAL REASONING FOR KNOWLEDGE-BASED VQA VIA DATA AND SAMPLING CURRICULUM

**Shan Ning**[1,3]**, Longtian Qiu**[1]**, Xuming He**[1,2]

[1]ShanghaiTech University, Shanghai, China
[2]Shanghai Engineering Research Center of Intelligent Vision and Imaging
[3]Lingang Laboratory, Shanghai, China
{ningshan2022,qiult,hexm}@shanghaitech.edu.cn

## ABSTRACT

Knowledge-Based Visual Question Answering (KB-VQA) requires models to answer questions about an image by integrating external knowledge, posing significant challenges due to noisy retrieval and the structured, encyclopedic nature of the knowledge base. These characteristics create a distributional gap from pretrained multimodal large language models (MLLMs), making effective reasoning and domain adaptation difficult in the post-training stage. In this work, we propose *Wiki-R1*, a data-generation-based curriculum reinforcement learning framework that systematically incentivizes reasoning in MLLMs for KB-VQA. Wiki-R1 constructs a sequence of training distributions aligned with the model's evolving capability, bridging the gap from pretraining to the KB-VQA target distribution. We introduce *controllable curriculum data generation*, which manipulates the retriever to produce samples at desired difficulty levels, and a *curriculum sampling strategy* that selects informative samples likely to yield non-zero advantages during RL updates. Sample difficulty is estimated using observed rewards and propagated to unobserved samples to guide learning. Experiments on two KB-VQA benchmarks, Encyclopedic VQA and InfoSeek, demonstrate that Wiki-R1 achieves new state-of-the-art results, improving accuracy from 35.5% to 37.1% on Encyclopedic VQA and from 40.1% to 44.1% on InfoSeek. The project page is available at https://artanic30.github.io/project_pages/WikiR1.

## 1 INTRODUCTION

Knowledge-Based Visual Question Answering (KB-VQA) is a challenging multimodal task that requires answering questions about an image by integrating external knowledge. A widely adopted approach is the Retrieval-Augmented Generation (RAG) framework, which leverages pretrained models and is further adapted to the task: a retriever first fetches relevant knowledge passages, and a generator then produces an answer conditioned on this context. However, the noise in the retrieval system is inherent, and the knowledge base (Vrandečić & Krötzsch, 2014) typically consists of structured, encyclopedic information. Consequently, the model must not only reason over noisy and imperfect external evidence but also comprehend retrieved information presented in a structured, encyclopedic form largely unseen during pretraining. These characteristics position KB-VQA as a challenging downstream task for pretrained MLLMs, one that demands robust reasoning ability and effective domain transfer, and is typically addressed in the post-training stage.

Prior work has pursued two main directions. One line aims to improve retrieval quality (Lerner et al., 2024; Yan & Xie, 2024; Yang et al., 2025; Deng et al., 2025), but retrieval remains inherently noisy and cannot guarantee full coverage of necessary evidence. Another line of work focuses on enhancing reasoning to handle imperfect retrieval. Specifically, models must understand encyclopedic passages and selectively extract relevant information while filtering out irrelevant content. Early efforts primarily relied on supervised fine-tuning (Caffagni et al., 2024; Qi et al., 2024; Cocchi et al., 2024), which enables models to reason over retrieved knowledge for specific training instances. However, our empirical results indicate that such approaches may have limited reasoning

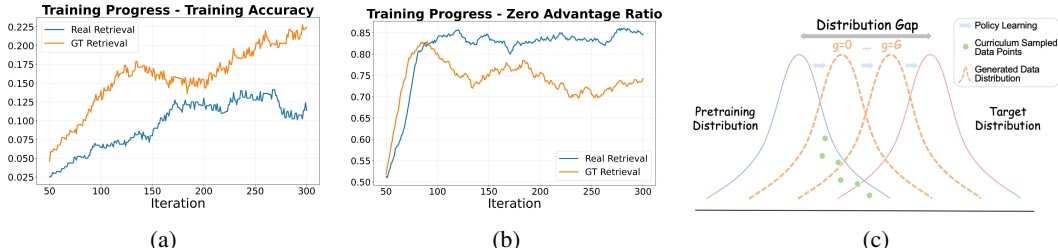

(a)          (b)          (c)

Figure 1: ( 1a) and ( 1b): **Training dynamics of DAPO on KB-VQA.** RL optimization suffers from a high proportion of zero-advantage samples and low training accuracy, highlighting the distribution gap between pretraining and the KB-VQA target domain. ( 1c): **Motivation of Wiki-R1.** To mitigate this gap, Wiki-R1 generates a sequence of training distributions with progressively reduced discrepancies and employs a curriculum sampling strategy to select informative samples.

robustness (Section 4.4). More recent reinforcement learning methods, including GRPO (Shao et al., 2024), have demonstrated promising reasoning capabilities in general retrieval-augmented generation (RAG) settings (Jin et al., 2025; Wu et al., 2025). Despite these advances, the effectiveness of RL-based approaches in tasks that require both multimodal reasoning and cross-domain adaptation, such as KB-VQA, remains largely unexplored.

To investigate this, we conduct preliminary experiments applying the popular RL algorithm DAPO (Yu et al., 2025) to incentivize the reasoning ability of MLLMs on KB-VQA. We observe that over 80% of the samples exhibit zero advantages(Figure 1a) during training, and the overall training accuracy remains low, around 10%(Figure 1b). These observations indicate that reinforcement learning on KB-VQA suffers from a severe sparse reward problem, which is exacerbated by the distributional gap between the model's pretraining data and the KB-VQA target domain. To further investigate the source of this distributional gap, we conduct experiments using the ground-truth retrieval, which corresponds to a setting with substantially reduced retrieval noise. As shown in Figure 1, both the prevalence of zero gradients and the low training accuracy are alleviated. This observation indicates that retrieval noise is a significant contributing factor to the sparse reward and ineffective training in RL for KB-VQA.

To address this challenge, we propose a data-generation-based curriculum reinforcement learning framework, *Wiki-R1*, designed to incentivize the reasoning ability of MLLMs on the challenging KB-VQA task. Wiki-R1 constructs a sequence of training distributions adaptively aligned with the model's evolving capability, gradually bridging the gap from pretraining to the KB-VQA target distribution, as illustrated in Figure 1. Unlike conventional curriculum learning, we generate training data with controllable difficulty rather than selecting from a fixed dataset. Specifically, we introduce *controllable curriculum data generation*, which manipulates the retriever to produce samples at the desired difficulty level, adaptively adjusted based on the model's observed training accuracy during RL optimization. Since generated data may not always match the intended difficulty, we further propose a *curriculum sampling strategy* that selects samples likely to yield non-zero advantages during RL updates. To estimate sample difficulty, we use observed rewards as a proxy and propagate this information to unobserved samples. Together, controllable data generation and curriculum sampling form a principled framework that systematically guides the model through progressively harder examples, ensuring meaningful learning signals and stable reinforcement learning on KB-VQA.

We evaluate our proposed framework on two standard knowledge-based visual question answering benchmarks: Encyclopedic-VQA (Mensink et al., 2023) and InfoSeek (Chen et al., 2023). Our method, Wiki-R1, achieves new state-of-the-art performance on both datasets, with an accuracy of 37.1% on Encyclopedic-VQA (surpassing the previous best of 35.5%) and 44.1% on InfoSeek (improving upon the prior state-of-the-art of 40.1%). Notably, on the challenging Unseen-Question split of InfoSeek, our model attains an accuracy of 47.8%. This performance not only exceeds the previous benchmark but also surpasses our model's overall accuracy, underscoring its strong generalization capability to novel queries.

Our main contributions are as follows:

- We propose *Wiki-R1*, a data-generation-based curriculum RL framework that incentivizes the reasoning ability of MLLMs on KB-VQA with data and sampling curriculum.

- Wiki-R1 constructs a curriculum of training distributions by manipulating the retrieval system and adaptively adjusting difficulty based on the model's performance. Curriculum sampling complements this process by selecting informative samples using propagated reward signals, ensuring the curriculum effectively guides learning.

- Experimental results demonstrate that Wiki-R1 consistently surpasses prior state-of-the-art methods on two challenging knowledge-based VQA benchmarks, with particularly pronounced improvements in unseen settings.

## 2 RELATED WORK

### 2.1 KNOWLEDGE-BASED VISUAL QUESTION ANSWERING

The KB-VQA task addresses questions whose answers require external or domain-specific knowledge beyond what is present in the image itself. Early datasets such as OK-VQA (Marino et al., 2019; Schwenk et al., 2022), FVQA (Wang et al., 2016), KVQA (Shah et al., 2019), S3VQA (Jain et al., 2021) and ViQuAE (Lerner et al., 2022a) posed questions requiring commonsense knowledge. Building on these datasets, a line of early methods (Gui et al., 2021; Marino et al., 2021; Hu et al., 2022; Ding et al., 2022; Lin & Byrne, 2022; Wu et al., 2022; Xenos et al., 2023) explored how to utilize external knowledge in VQA through structured knowledge graphs, multi-modal reasoning, or evidence retrieval. With the emergence of LLM-based MLLMs, these early datasets provide only limited coverage for evaluating KB-VQA in more realistic scenarios, as they often lack fine-grained knowledge, require minimal visual understanding, and cover only a restricted range of visual entity categories, as noted by (Chen et al., 2023). To address these limitations, recent benchmarks such as Encyclopedic-VQA (Mensink et al., 2023) and InfoSeek (Chen et al., 2023) present greater challenges by targeting highly specific, Wikipedia-scale knowledge. They require models to capture detailed information about particular entities and nuanced encyclopedic facts.

To tackle this task, Retrieval-Augmented Generation (RAG) has emerged as a widely adopted paradigm, where models retrieve relevant content from external knowledge bases such as Wikipedia to support question answering. Recent studies can be broadly categorized into two directions. The first focuses on improving the retrieval system itself, for instance, by training contrastive image-text encoders to achieve more accurate retrieval results (Xu et al., 2024; Radford et al., 2021; Sun et al., 2023; Wei et al., 2023; Xiao et al., 2024; Caffagni et al., 2024; Ning et al., 2023; Qiu et al., 2024). However, due to the large scale of knowledge bases and the inherent long-tail distribution of training data, retrieval noise is often unavoidable. The second line of work, therefore, aims to adapt models to noisy retrieval outputs. For example, Wiki-LLaVA (Caffagni et al., 2024) integrates external multimodal knowledge via a hierarchical retrieval pipeline within a contrastive embedding space (Radford et al., 2021). RoRAVLM (Qi et al., 2024) instead introduces a visual token refinement module to filter out query-irrelevant visual information from both retrieved and query images. More recently, ReflectiVA (Cocchi et al., 2024) employs reflective tokens to dynamically determine the reliability of retrieved content, thereby mitigating the impact of noisy retrieval results. In this work, we propose to leverage reinforcement learning to enhance the model's ability to reason under noisy retrieval conditions, enabling it to derive correct answers even when the retrieved content is imperfect.

### 2.2 CURRICULUM LEARNING FOR RL

Curriculum learning (Bengio et al., 2009; Graves et al., 2017) structures the training process by gradually moving from easier to more difficult examples. In reinforcement learning, curricula are typically based on task complexity (Justesen et al., 2018; Wang et al., 2019; Li et al., 2019), or alternatively learned through teacher–student frameworks formulated as partially observable Markov decision processes (Matiisen et al., 2017; Portelas et al., 2019). With the success of DeepSeek-R1, recent studies have explored incorporating curriculum learning into value-free RL frameworks such as GRPO (Shao et al., 2024; Yu et al., 2025; Qiu et al., 2025; 2026). For instance, ADARFT (Shi et al., 2025a) dynamically prioritizes samples with higher learning potential based on recent reward signals, while DUMP (Wang et al., 2025b) adopts the Upper Confidence Bound principle to adaptively adjust sampling probabilities across different data distributions. In the context of multimodal RAG, several works (Ji et al., 2025; Wang et al., 2025a; Zhang et al., 2025) apply fixed curric-

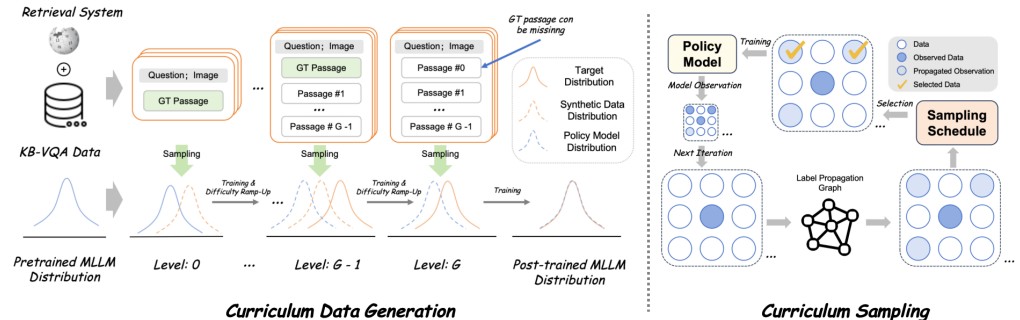

Figure 2: **Left: Controllable curriculum data generation.** We manipulate the retriever to generate training samples with gradually increasing difficulty, adaptively aligned with the model's evolving capability, bridging the gap from pretraining to the KB-VQA target distribution. **Right: Curriculum sampling with observation propagation.** We adaptively select informative samples likely to produce non-zero advantage during RL updates, with sample difficulty estimated from observed rewards and propagated to unobserved examples.

ula, training policies progressively from easy to hard samples. More advanced approaches, such as VL-Cogito (Yuan et al., 2025), estimate sample difficulty using current reward signals and dynamically adjust sample weights accordingly. In this work, we go beyond selection-based curricula and instead generate controllable training distributions, enabling principled, difficulty-aware data construction that bridges the gap between pretraining and target distribution. We further introduce an observation-propagation mechanism that propagates sparse on-policy reward signals to unobserved examples, yielding reliable difficulty estimates to drive curriculum sampling.

## 3  WIKI-R1

In this section, we present *Wiki-R1*, a curriculum reinforcement learning framework that enhances the reasoning ability of multimodal large language models (MLLMs) for the challenging knowledge-based visual question answering (KB-VQA) task. We first formulate the KB-VQA task to establish the problem setting in Section 3.1, and then introduce the training objective under our post-training reinforcement learning setting in Section 3.2. Building on this objective, we design two tightly coupled components: (i) *curriculum data generation* (Section 3.3), which constructs a progressive training sequence from easy to hard by manipulating the retrieval system, and (ii) *curriculum sampling* (Section 3.4), which dynamically selects informative samples via observation propagation. The overall pipeline is illustrated in Figure 2, with a detailed pseudo-code provided in the appendix.

### 3.1  TASK DEFINITION

The goal of Knowledge-based Visual Question Answering (KB-VQA) is to generate an answer $y$ to a textual query $q$ given an image $I^q$, by jointly reasoning over the input and relevant knowledge retrieved from an external knowledge base (KB). Formally, we define a large-scale multimodal KB, such as Wikipedia (Vrandečić & Krötzsch, 2014), as

$$\mathcal{B} = (P_i, I_i)_{i=1}^{N}, \tag{1}$$

where $P_i$ denotes a textual article and $I_i$ is the corresponding visual content associated with entity $i$. To incorporate external knowledge, a retriever is employed to select a subset of relevant multimodal documents, and a non-parametric, rule-based retrieval modification function is also incorporated to adjust the retrieval results:

$$S_\phi = \text{Retriever}(q, I^q, \mathcal{B}), \quad S \subset \mathcal{B}, \tag{2}$$

where $\phi$ denotes the retrieval modification function and $S$ contains the knowledge passages most relevant to the query $(q, I^q)$. The retrieved set $S_\phi$ serves as additional context for answer generation. Formally, the KB-VQA objective is to model the conditional distribution of the answer $y$ given the query $q$, the image $I^q$, and the retrieved knowledge $S_\phi$:

$$\max_{\theta} \ \mathbb{E}_{(I^q,q,y)\sim\mathcal{D}}\Big[\log p_\theta(y \mid I^q, q, S_\phi)\Big], \tag{3}$$

where $\theta$ denotes the learnable parameters and $\mathcal{D}$ is the KB-VQA dataset.

## 3.2 Training Objective

To address the challenging KB-VQA task that requires reasoning ability, we consider a post-training setting in which a pretrained MLLM is further adapted to the KB-VQA task via reinforcement learning. In this stage, the model is optimized to maximize the expected reward over KB-VQA data, conditioned on the query–image pair $(q, I^q)$ and the retrieved knowledge $S_\phi$. A key challenge is the sparsity of reward signals, which can hinder stable optimization. To address this, we leverage the retrieval modification function $\phi$ and further introduce a sampling schedule $\mu$, which together shape the learning signal and training distribution. Formally, the gradient under $(\mu, \phi)$ for the model policy $\pi_\theta$ is given by

$$\nabla_\theta J(\pi_\theta, \mu, \phi) = \mathbb{E}_{(q, I^q, y) \sim \mu} \ \mathbb{E}_{\hat{y} \sim \pi_\theta(\cdot | q, I^q, S_\phi)} \Big[ \nabla_\theta \log \pi_\theta(\hat{y} \mid q, I^q, S_\phi) \, r(\hat{y}, y) \Big]. \tag{4}$$

where $\hat{y}$ is the sampled answer from the policy $\pi_\theta$, and $y$ denotes the ground-truth answer.

Unlike traditional policy gradient methods that rely on random sampling for $\mu$ and a fixed retrieval strategy for $\phi$, our framework *Wiki-R1* explicitly incorporates curriculum-aware sampling and controllable retrieval modifications. This design provides a principled way to align data generation with optimization, thereby mitigating the gap between pretraining and target distributions. Further details are presented in the following sections.

## 3.3 Curriculum Data Generation

**Controllable Data Generation**   To systematically bridge the gap from pretraining to the KB-VQA target distribution, we manipulate the retriever to generate a sequence of training samples with controllable difficulty. Intuitively, retrieving more candidates increases the likelihood of including useful query-relevant information, but also introduces additional noise. Motivated by this property, we design a controllable data generation method that adjusts both the number of retrieved candidates and whether the ground-truth article is explicitly included in the retrieval results, which is illustrated in Figure 2.

Specifically, we define a discrete gap level $g \in \{0, 1, \ldots, G\}$, which represents the degree of distribution shift between the generated training samples and the true KB-VQA target distribution. For each level $g$, we instantiate a retrieval modification function $\phi_g(k, \gamma)$ that specifies the number of retrieved candidates $k$ and whether the ground-truth snippet $\gamma$ is enforced.

- **Easiest level** ($g = 0$): we set $k = 1$ and $\gamma = 1$, which retrieve only the ground-truth snippet.
- **Intermediate levels** ($1 < g < G$): $k$ is set to $g$ while keeping $\gamma = 1$, introducing noisy candidates alongside the ground truth.
- **Hardest level** ($g = G$): set $\gamma = 0$ and $k = G - 1$, so the retrieval system no longer guarantees inclusion of the ground truth, fully aligning with the inference-time distribution.

This design produces a controllable hierarchy of training distributions, beginning with $g = 0$, which closely resembles the pretraining distribution, and gradually converging to the target KB-VQA distribution at $g = G$.

**Gap-Level Schedule**   To dynamically adjust the gap level during training, we design a schedule based on the model's observed training accuracy. Concretely, we maintain a sliding window of the most recent $w$ samples and compute the average training accuracy. Once this moving average exceeds an upgrade threshold $\tau$, we promote the gap level $g \mapsto g + 1$ and reset the stored observations. This mechanism ensures that the model is gradually exposed to more challenging training distributions only after it has sufficiently mastered the current level, enabling a smooth transition from pretraining-like data to the target KB-VQA distribution.

## 3.4 Curriculum Sampling with Observation Propagation

**Sampling Schedule**   The training data generated by our controllable curriculum may not fully satisfy the desired difficulty. To address this, we introduce a curriculum sampling strategy $\mu$. Prior work (Shi et al., 2025b) has shown that samples with a training accuracy near $0.5$ provide the strongest gradient signal for reinforcement learning. Accordingly, during training, we sample data using a Gaussian distribution centered at the historical mean training accuracy of $0.5$.

Formally, we denote by $\mu$ a sampling schedule represented as a distribution over $\mathcal{D}$:

$$(q, I^q, y) \sim \mu(\cdot), \quad \mu \in \Delta(\mathcal{D}), \tag{5}$$

where $(q, I^q, y)$ denotes a sampled training data. This ensures that the model primarily trains on samples that are challenging yet solvable, maximizing learning efficiency and stabilizing the RL optimization process.

**Difficulty Estimation via Observation Propagation**   A key challenge in the sampling schedule is the sample difficulty estimation. Though observed reward provides a direct evaluation of data, it is extremely sparse, which can undermine the effectiveness of curriculum sampling. To address this, we introduce an *observation propagation* mechanism to estimate the difficulty of unobserved samples, which is illustrated in Figure 2. We leverage the insight that the correlation between different VQA samples is related to the model's understanding of their associated knowledge base article. Concretely, we construct a label propagation graph over VQA samples, where the edge weights between two samples reflect the similarity of their associated knowledge base articles. We then apply label propagation to propagate observed accuracies from the training set to unobserved samples. This allows us to approximate sample-wise expected accuracies, ensuring that curriculum sampling remains effective even under sparse observations. We provide the details of label propagation in the appendix.

## 4 EXPERIMENTS

In this section, we present the experimental validation of our method on two challenging benchmarks, along with the implementation details. Moreover, we conduct comprehensive ablation studies to demonstrate the effectiveness of each key component of our method.

### 4.1 EVALUATION BENCHMARKS

**Encyclopedic VQA.**   To evaluate the performance of multi-modal large language models (MLLMs) on visual questions requiring extensive external knowledge, we utilize the recently proposed Encyclopedic VQA (EVQA) (Mensink et al., 2023) dataset. This dataset contains visual questions about detailed properties of fine-grained categories and is primarily constructed using annotations from iNaturalist 2021 (Horn et al., 2021) and the Google Landmarks Dataset V2 (Weyand et al., 2020). The Encyclopedic VQA dataset comprises approximately 221k question-answer pairs associated with 16.7k different fine-grained entities, each represented by up to five images. The dataset is divided into training, validation, and test splits, containing 1M, 13.6k, and 5.4k samples, respectively. For the knowledge base, Encyclopedic VQA filters out non-English Wikipedia pages from the WIT dataset (Srinivasan et al., 2021) and compiles a total of 2M Wikipedia pages. We report the BEM (Bulian et al., 2022) score of the test set using official scripts.

**InfoSeek.**   The InfoSeek (Chen et al., 2023) benchmark is tailored for information-seeking questions that require expert knowledge. It consists of 1.3 million visual information-seeking questions, encompassing more than 11,000 visual entities from OVEN (Hu et al., 2023). The dataset comprises 934k training, 73k validation, and 348k test samples. Due to computational resource restrictions, we sample a class-balanced 1k validation set to report the final performance with official scripts and select another 1k subset for hyperparameter selection. For the knowledge base, we follow previous works (Yan & Xie, 2024; Cocchi et al., 2024) and utilize a knowledge base with 100,000 Wikipedia articles accompanied by images.

### 4.2 BASELINES

To evaluate the effectiveness of our method, we consider two categories of baselines. *(1) Zero-shot MLLMs.* The first category consists of zero-shot multimodal large language models (MLLMs). We evaluate models of different scales, including BLIP-2 (Li et al., 2023), InstructBLIP (Dai et al., 2023), LLaVA 1.5 (Liu et al., 2023), Qwen2.5-VL (Bai et al., 2025), and GPT-4V (OpenAI, 2023). These models are directly applied to KB-VQA without retrieval augmentation, which highlights the inherent difficulty of the task. *(2) Retrieval-augmented Generation.* The second category corresponds to methods under the retrieval-augmented generation (RAG) setting. In this setting, models enhance answer accuracy by retrieving relevant snippets from an external knowledge base. Since our focus is on KB-VQA with a noisy retrieval system, we primarily compare with methods that do

Table 1: **Performance comparison on Encyclopedic VQA and InfoSeek.** All results of retrieval-augmented generation methods are reported without applying any re-ranking stage to reorder retrieved documents. *Retrieval Mode* spans two columns: the first specifies the retrieval model, while the second indicates the type of knowledge source utilized. The *V.* and *T.* indicate the visual and textual retrieval mode. The *Con.* and *Col.* indicate textual retrieval model, Contriver (Izacard et al., 2021) and Colbert V2 (Santhanam et al., 2021) respectively.

| Method | Retrieval Mode | | EVQA | | InfoSeek | | | Avg. |
|---|---|---|---|---|---|---|---|---|
| | | | Single-hop | All | Unseen-Q | Unseen-E | All | |
| *Zero-shot MLLMs* | | | | | | | | |
| BLIP-2 | - | - | 12.6 | 12.4 | 12.7 | 12.3 | 12.5 | 12.5 |
| InstructBLIP | - | - | 11.9 | 12.0 | 8.9 | 7.4 | 8.1 | 10.1 |
| LLaVA-1.5 7B | - | - | 16.0 | 16.9 | 8.3 | 8.9 | 7.8 | 12.4 |
| Qwen-2.5-VL 3B | - | - | 18.6 | 18.8 | 26.3 | 16.1 | 19.6 | 19.2 |
| Qwen-2.5-VL 7B | - | - | 26.6 | 26.3 | 25.3 | 17.2 | 19.9 | 23.1 |
| GPT-4V | - | - | 26.9 | 28.1 | 15.0 | 14.3 | 14.6 | 21.4 |
| *Retrieval-Augmented Generation* | | | | | | | | |
| DPR$_{V+T}$ | CLIP ViT-B/32 | V. + T. | 29.1 | - | - | - | 12.4 | - |
| RORA-VLM | CLIP+Google Search | V. + T. | - | 20.3 | 25.1 | 27.3 | - | - |
| Wiki-LLaVA | CLIP ViT-L/14+Con. | T. | 18.3 | 19.6 | 28.6 | 25.7 | 27.1 | 23.4 |
| EchoSight | EVA-CLIP-8B | T. | 22.4 | 21.7 | 30.0 | 30.7 | 30.4 | 26.1 |
| EchoSight | EVA-CLIP-8B | V. | 26.4 | 24.9 | 18.0 | 19.8 | 18.8 | 21.9 |
| ReflectiVA | CLIP ViT-L/14 | T. | 24.9 | 26.7 | 34.5 | 32.9 | 33.7 | 30.2 |
| ReflectiVA | EVA-CLIP-8B | T. | 28.0 | 29.2 | 40.4 | 39.8 | 40.1 | 34.7 |
| ReflectiVA | EVA-CLIP-8B | V. | 35.5 | 35.5 | 28.6 | 28.1 | 28.3 | 31.9 |
| Wiki-R1 3B | EVA-CLIP-8B + Col. | V.+ T. | 40.4 | 35.9 | 46.0 | 40.3 | 42.2 | 39.1 |
| Wiki-R1 7B | EVA-CLIP-8B + Col. | V.+ T. | **41.0** | **37.1** | **47.8** | **42.3** | **44.1** | **40.6** |

not perform dedicated retriever training, including DPR (Lerner et al., 2024), RORA-VLM (Qi et al., 2024), Wiki-LLaVA (Caffagni et al., 2024), EchoSight (Yan & Xie, 2024), and ReflectiVA (Cocchi et al., 2024).

Table 2: **Performance comparison on the ViQuAE benchmark under zero-shot transfer setting.** We compare against RC baselines (Lerner et al., 2022a) and MLLM methods. Wiki-R1 consistently outperforms all prior methods, even surpassing the RC semi-oracle configuration.

| Type | Model | F1 | Exact Match |
|---|---|---|---|
| RC Baseline | Zero-shot | 20.96 | 18.06 |
| | Few-shot | 25.43 | 22.07 |
| | Few-shot (semi-oracle) | 44.10 | 40.32 |
| | Few-shot (full-oracle) | 63.17 | 57.55 |
| MLLM Baseline | LLaVA-v1.5 | 15.1 | 26.6 |
| | Wiki-LLaVA (E-VQA) | 10.5 | 16.7 |
| | Wiki-LLaVA (InfoSeek) | 12.7 | 21.8 |
| | ReflectiVA | 23.2 | 38.1 |
| Ours | Wiki-R1 3B | 53.8 | 48.6 |
| | Wiki-R1 7B | 55.6 | 50.3 |

## 4.3 IMPLEMENTATION DETAILS

**Training Data.** To implement reinforcement learning under the KB-VQA setting, we construct a balanced training set by sampling examples according to their ground-truth entities. Specifically, we construct entity-balanced subsets by sampling 20k examples from Encyclopedic VQA (Mensink et al., 2023) and 20k examples from InfoSeek (Chen et al., 2023), ensuring that each entity is proportionally represented within the subsets. The resulting training set contains a total of 40k examples. Notably, the scale of our training data is far smaller compared with baselines, and we provide a data scale comparison in the appendix.

Table 3: **Results under the oracle Wikipedia entity setting.** *KB Article* denotes providing the entire ground-truth Wikipedia article to the MLLM, while *KB Passage* denotes using model-specific strategies to retrieve relevant passages within the article. The LLM column specifies the large language model backbone used in each method.

| Method | LLM | EVQA Single-hop | InfoSeek Unseen-Q | Unseen-E | Overall |
|--------|-----|-----------------|-------------------|----------|---------|
| *KB Article* | | | | | |
| LLaVA-v1.5 | Vicuna-7B | 42.9 | 14.2 | 13.4 | 13.8 |
| LLaVA-v1.5 | LLaMA-3.1-8B | 54.1 | 20.1 | 17.7 | 18.8 |
| *KB Passage* | | | | | |
| Wiki-LLaVA | LLaMA-3.1-8B | 46.8 | 51.2 | 50.6 | 50.9 |
| ReflectiVA | LLaMA-3.1-8B | **75.2** | 57.8 | 57.4 | 57.6 |
| Wiki-R1(Ours) | Qwen-2.5-3B | 68.5 | 64.0 | 65.9 | 65.3 |
| Wiki-R1(Ours) | Qwen-2.5-7B | 69.2 | **65.5** | **69.5** | **68.2** |

**Training Details.** We adopt the widely used VERL (Volcengine, 2025) framework and implement our proposed design based on the DAPO (Yu et al., 2025) algorithm. The reward function for reinforcement learning is defined as a rule-based binary signal: the model receives a reward of 1 if the generated answer exactly matches the ground-truth answer, and 0 otherwise. The learning rate for both variants is set to 1e-6, and we set the number of rollouts for each sample to 4. For other hyperparameters, we follow the official scripts. For base models, we employ the recently released Qwen2.5-VL (Bai et al., 2025) models (3B and 7B), which represent the state-of-the-art among open-source multimodal language models. For curriculum data generation, the window size $w$ is set to 300, the gap threshold $\tau$ is 0.55, and the maximum gap $G$ is set to 6. The training takes 9 hours for the 3B variant and 12 hours for the 7B variant on 4 A100 GPUs.

**Retrieval System** We follow previous works (Yan & Xie, 2024) that utilize EVA-CLIP 8B to compute the visual similarity score and utilize ColBERT V2 (Santhanam et al., 2021) to extract the relevant text chunks and compute the question relevance score. We use a weighted sum to combine these scores. The score weight is selected based on the recall on the training set of Encyclopedic VQA and InfoSeek, respectively. More details are provided in the appendix.

## 4.4 PERFORMANCE ANALYSIS

**Comparison with State of Art.** We evaluate our model on the two benchmarks described above, comparing against zero-shot multimodal LLMs (MLLMs), and retrieval-augmented baselines. As shown in Table 1, our proposed method with the 3B variant surpasses previous state-of-the-art approaches. Moreover, our framework *consistently achieves strong performance across both benchmarks using a single retrieval system*, in contrast to prior methods such as EchoSight and ReflectiVA, whose performance is highly sensitive to the retrieval mode. For instance, ReflectiVA (Cocchi et al., 2024) attains 35.5 on EVQA under visual retrieval, but its accuracy on InfoSeek drops to 28.3 compared to 40.1 with textual retrieval. These results demonstrate that our framework is not only more robust across benchmarks but also achieves superior overall performance.

**Inference with Oracle Documents.** To comprehensively evaluate our model, we further conduct experiments under an *oracle* setting, where the ground-truth entity (i.e., the Wikipedia page associated with the query) is directly provided. In this configuration, Wiki-R1 is only given retrieval results from the ground-truth entity, while the passages within the article may still contain noise. Thus, this setting can be regarded as the upper bound of our approach by eliminating entity-level retrieval noise. As shown in Table 3, Wiki-R1 shows strong performance on both benchmarks, demonstrating its strong ability to effectively leverage correct retrieval results.

**Generalization to ViQuAE** To further assess the generalization capability of Wiki-R1, we conduct zero-shot transfer evaluation on the ViQuAE benchmark (Lerner et al., 2022b). Following the official evaluation protocol, we report F1 and Exact Match scores using the provided 1.4M Wikipedia article knowledge base. As shown in Table 2, Wiki-R1 achieves 53.8 F1 and 48.6 EM with the 3B model, and 55.6 F1 and 50.3 EM with the 7B model. Our method substantially outperforms existing MLLM baselines such as ReflectiVA and even surpasses the RC semi-oracle set-

Table 4: **Ablation study of framework design on Encyclopedic VQA and InfoSeek.** We conduct experiments on Qwen-2.5-VL 3B model. Each row progressively adds components, and we mark enabled modules with ✓. The *Samp. Cur., Data Cur., Obs. Prop.* indicate the sampling curriculum, data curriculum generation, and observation propagation strategies.

| Method | Modules | | | EVQA | | InfoSeek | | |
| | Data Cur. | Samp. Cur. | Obs. Prop. | Single-hop | Overall | Unseen-Q | Unseen-E | Overall |
| --- | --- | --- | --- | --- | --- | --- | --- | --- |
| Zero-shot | - | - | - | 18.6 | 18.8 | 26.3 | 16.1 | 19.6 |
| SFT | - | - | - | 21.6 | 25.1 | 38.7 | 24.9 | 29.5 |
| SFT | ✓ | - | - | - | 34.4 | - | - | 32.1 |
| DAPO | × | × | × | 35.9 | 31.4 | 44.9 | 39.8 | 41.5 |
| | ✓ | × | × | 39.4 | 34.5 | **46.9** | **41.1** | **43.0** |
| | ✓ | ✓ | × | 36.4 | 32.1 | 45.2 | 37.3 | 40.0 |
| | ✓ | ✓ | ✓ | **40.4** | **35.9** | 46.0 | 40.3 | 42.2 |

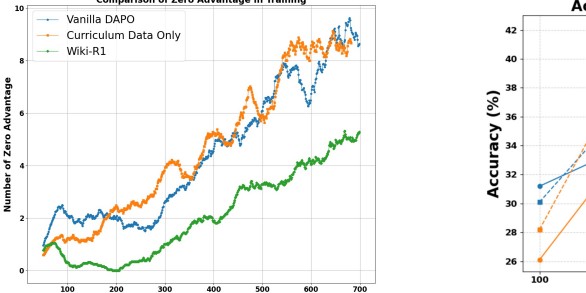 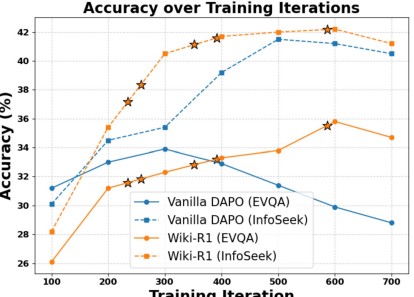

Figure 3: **Left: Number of ignored trajectories.** Trajectories are ignored when they provide zero advantage and no training signal; a larger number indicates lower training efficiency. **Right: Accuracy over training iterations.** Performance is reported on the EVQA test set and the InfoSeek validation set. The *star* denotes an increase in curriculum difficulty during Wiki-R1 training.

ting. These results demonstrate that Wiki-R1 exhibits strong cross-dataset generalization to unseen knowledge sources and question distributions.

### 4.5 Ablation Study

**Effectiveness of Curriculum Data Generation** To assess the contribution of each component in our framework, we conduct a detailed ablation study. We start from a supervised fine-tuning (SFT) baseline, and then incorporate the strong reinforcement learning algorithm DAPO (Yu et al., 2025). Building upon DAPO, we further introduce a curriculum data generation strategy, which adapts the retrieval policy to construct training data from easier to more challenging instances.

As shown in Table 4, naive SFT yields only limited improvements, while DAPO, as a powerful RL algorithm, achieves substantial gains. Our proposed data curriculum further enhances the effectiveness of DAPO, particularly on the more challenging EVQA benchmark, highlighting the importance of curriculum-guided data generation in noisy retrieval settings.

**Effectiveness of Curriculum Sampling** We further analyze the proposed sampling strategy by introducing curriculum sampling on top of data curriculum, and then augmenting it with observation propagation. As shown in Table 4, directly applying curriculum sampling alone leads to degraded performance. We attribute this to the sparsity of observations: selecting the next training stage solely based on the accuracy of observed samples tends to either repeatedly select a small subset of seen samples or randomly sample from entirely unobserved instances. This highlights the necessity of our observation propagation module, which alleviates the sparsity issue and enables curriculum sampling to function as intended, thereby improving both training efficiency and effectiveness.

**Efficiency of Observation Propagation** Our proposed observation propagation module addresses the sparsity of observations by efficiently identifying samples required for constructing the sampling curriculum. This reduces the number of skipped trajectories that contain no reward signal. To illustrate this effect, we compare three settings: (i) *Vanilla DAPO*, (ii) DAPO with a curriculum sampling

schedule, denoted as *Curriculum Sampling Only*, and (iii) DAPO with curriculum sampling plus our observation propagation, resulting in the *Wiki-R1*. As shown in Figure 3, observation propagation significantly decreases the number of skipped trajectories during training, thereby improving the efficiency of RL optimization. Moreover, by reducing wasted samples, it simultaneously enhances the overall training effectiveness.

**Visualization of Training Dynamics.** To gain deeper insights into the behavior of our framework, we track the performance of DAPO and Wiki-R1 across training iterations. As shown in Figure 3, DAPO exhibits rapid improvement in the early stage (e.g., within the first 100 iterations), but its performance on EVQA degrades as training progresses. We attribute this to overfitting on the relatively easier InfoSeek dataset: compared to InfoSeek, EVQA involves noisier retrieval results (Table 6), which deviate further from the MLLM's pretrained distribution. In contrast, Wiki-R1 with curriculum training achieves stable improvements on both benchmarks, and its best performance emerges when training reaches the highest curriculum difficulty level—closely matching the challenges in real inference scenarios.

**Similarity Measures on Label Propagation** To assess the impact of the similarity measure used in label propagation, we compare TF-IDF with a semantically-aware text embedding model (Sentence Transformer) in Table 5. The results demonstrate that the proposed framework generalizes across different similarity metrics and benefits from semantically enriched representations.

Table 5: Different similarity measures for label propagation.

| Method | EVQA | | InfoSeek | | | Avg. |
|---|---|---|---|---|---|---|
| | Single-hop | Overall | Unseen-Q | Unseen-E | Overall | |
| TF-IDF | 40.4 | 35.9 | 46.0 | 40.3 | 42.2 | 39.1 |
| Sentence Transformer | 41.1 | 36.3 | 46.2 | 41.3 | 43.0 | 40.7 |

## 5 LIMITATION

While our proposed Wiki-R1 effectively incentivizes the reasoning ability of MLLMs on KB-VQA, it also has certain limitations. In particular, manipulating the retrieval system provides only a partial means of controlling the gap between the pretraining and target distributions, rather than a fully controllable data generation process. We view this as a promising direction for future research, where advances in controllable data generation could enable more principled curriculum design for KB-VQA and related tasks.

## 6 CONCLUSION

In this work, we introduce Wiki-R1, a data-generation-based curriculum reinforcement learning framework that incentivizes the reasoning ability of multimodal large language models on challenging KB-VQA tasks. By constructing a sequence of training distributions aligned with the model's evolving capability, and combining controllable curriculum data generation with adaptive curriculum sampling, Wiki-R1 effectively mitigates sparse reward issues and guides the model through progressively harder examples. Extensive experiments on Encyclopedic VQA and InfoSeek demonstrate significant improvements over state-of-the-art methods, including strong generalization to unseen questions. Our framework provides a principled approach for integrating retrieval and reinforcement learning in downstream tasks with distributional gaps, offering insights for future research on domain-adaptive reasoning in retrieval-augmented multimodal settings.

## 7 ACKNOWLEDGMENTS

This work was supported by NSFC 62350610269, Shanghai Frontiers Science Center of Human-centered Artificial Intelligence, and MoE Key Lab of Intelligent Perception and Human-Machine Collaboration (ShanghaiTech University). This work was also supported by HPC platform of ShanghaiTech University.

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

Table 6: **Retrieval results on EVQA test and InfoSeek validation sets.** We report Recall@K for $K = \{1, 5, 10, 20\}$. The CLIP I-I is the retrieval with the visual similarity score from EVQA-CLIP 8B only.

| Methods | Retrieval Mode | EVQA Test | | | | InfoSeek Val | | | |
|---|---|---|---|---|---|---|---|---|---|
| | | R@1 | R@5 | R@10 | R@20 | R@1 | R@5 | R@10 | R@20 |
| CLIP I-I | Visual | 11.0 | 26.2 | 33.8 | 41.0 | 45.7 | 65.7 | 71.6 | 76.2 |
| ReflectiVA | Textual | 10.1 | 20.5 | - | 29.4 | **56.1** | **77.6** | - | **86.4** |
| ReflectiVA | Visual | 15.6 | 36.1 | - | **49.8** | 29.6 | 41.4 | - | 46.6 |
| Wiki-R1 | Visual+Textual | **16.7** | **41.0** | **44.8** | 47.5 | 46.9 | 67.1 | **72.9** | 77.2 |

## A  APPENDIX

### A.1  DETAILS OF RETRIEVAL SYSTEM

In this section, we provide a detailed design of our retrieval system, which consists of two main modules: a *visual-based retrieval* module and a *textual-based retrieval* module. To combine the outputs of these two modules, we employ a *score fusion* strategy that weights and merges the visual and textual retrieval scores to produce a final ranking of candidate knowledge snippets for each query. The performance is shown in Table 6

**Visual-based Retrieval**  We first perform a coarse-level retrieval using a visual-based approach. Following previous work (Yan & Xie, 2024; Cocchi et al., 2024), we employ EVA-CLIP 8B (Sun et al., 2023) to extract global visual features from the query image $I^q$ and the images $I$ in the knowledge base $\mathcal{B}$. The similarity between the query and candidate images is then computed using the cosine similarity of their corresponding feature vectors. This provides an initial ranking of candidate knowledge items based on visual relevance.

**Textual-based Retrieval**  In the textual-based retrieval stage, we aim to achieve two objectives: (i) extract query-relevant textual passages from each knowledge base article, and (ii) assess the relevance of the article to the query using a text retrieval model. Specifically, we employ the ColBERT V2 (Santhanam et al., 2021) model and split each article into chunks of size 256. The relevance score of an article to a given query $q$ is determined by the highest relevance score among its retrieved passages.

**Retrieval Score Fusing**  After obtaining the visual similarity score $V$ and textual relevance score $T$ for each knowledge base article, we fuse the two scores using a weighted sum:

$$s_r = \lambda \cdot V + (1 - \lambda) \cdot T, \tag{6}$$

where $\lambda \in [0, 1]$ is a tunable hyperparameter controlling the relative importance of visual and textual cues. We select $\lambda$ based on the training set: $\lambda = 0.985$ for EVQA and $\lambda = 0.997$ for InfoSeek. The values are close to 1 because $V$ is normalized to $V \in [0, 1]$ while $T$ is unnormalized and can take values $T \in [0, +\infty)$.

### A.2  PSEUDO CODE FOR WIKI-R1

To provide a clearer overview of the training process, we present the pseudo code of *Wiki-R1* in Algorithm 1.

### A.3  TRAINING DATA SCALE COMPARISON

In this section, we provide a comparison of the training data scale between our proposed framework and baseline methods. As shown in Table 7, our method requires substantially fewer training samples while achieving superior performance. This highlights the efficiency of Wiki-R1 and demonstrates its applicability in scenarios with limited computational or data resources.

---

**Algorithm 1** Wiki-R1 - Data and Sampling Curriculum Reinforcement Learning

---

**Require:** KB-VQA Training dataset $\mathcal{D}$, knowledge base $\mathcal{B}$, policy model $\pi_\theta$, gap threshold $\tau$, reward function $r(\cdot, \cdot)$, Propagation Graph $\mathcal{K}$

1: Select RL algorithm $\mathcal{A}$ (e.g., PPO,GRPO,DAPO)
2: Initialize retrieval function $\phi_g$ where $g = 1$
3: Initialize estimated sample reward $\mathcal{H} \leftarrow \{0\}^{|\mathcal{D}|}$
4: Initialize sliding window reward $\mathcal{W} \leftarrow []$
5: **while** training is not finished **do**
6:     Select sample with target difficulty $(q, I^q, y^*)$ from $\mathcal{D}$ according to $\mathcal{H}$    ▷ Curriculum sampling with predicted sample reward
7:     $S \leftarrow \text{Retriever}(q, I^q, \mathcal{B}; \phi_g)$    ▷ Controllable Data Generation
8:     Construct Batch Data $X \leftarrow (q, I^q, S)$
9:     Generate responses $G = \pi_\theta(X)$
10:    Compute reward: $R \leftarrow r(X, G)$
11:    Update Policy $\theta \leftarrow \mathcal{A}(\pi_\theta, X, G, R)$
12:    Maintain sliding window $\mathcal{W} \leftarrow (\mathcal{W} \cup \{R\})[-w:]$    ▷ Only preserve last $w$ elements
13:    **if** $\frac{1}{|\mathcal{W}|} \sum_{w_i \in \mathcal{W}} w_i \geq \tau$ and $g < G$ **then**    ▷ Upgrade retrieval modification function
14:       $g \leftarrow g + 1$
15:       Reset $\mathcal{W} \leftarrow []$
16:    **end if**
17:    $\tilde{\mathcal{H}} \leftarrow \text{Propagate}(\{R\}, \mathcal{K})$    ▷ Difficulty estimation via observation propagation
18:    **for** each index $i$ with $\tilde{\mathcal{H}}[i] > 0$ **do**
19:       $\mathcal{H}[i] \leftarrow \mathcal{H}[i] + \frac{1}{2} * \tilde{\mathcal{H}}[i]$
20:    **end for**
21: **end while**

---

Table 7: Comparison of training data scale and performance across different methods.

| Method | FT Retrieval | FT Generation | EVQA | InfoSeek |
|---|---|---|---|---|
| Wiki-LLaVA | $\times$ | $\checkmark$ | 916,385 | 902,509 |
| Echosight | $\checkmark$ | $\times$ | 916,385 | 902,509 |
| ReflectiVA | $\times$ | $\checkmark$ | 2,900,000 | 2,500,000 |
| Wiki-R1 | $\times$ | $\checkmark$ | 20,000 | 20,000 |

## A.4 DETAILS OF OBSERVATION PROPAGATION

In this section, we provide a detailed design of the *observation propagation* mechanism used in curriculum sampling. The goal of this component is to estimate the difficulty of unobserved training samples by propagating the limited reward signals observed during RL training. By leveraging correlations among VQA samples that share the same knowledge base article, we can predict the expected reward for unobserved samples, enabling more effective curriculum-based difficulty estimation and sample selection.

**Graph Construction** To implement observation propagation, we first model the correlations between VQA samples as a label propagation graph $K$. Specifically, the correlation between samples is derived from the associated ground-truth knowledge base articles. To quantify the relatedness between different articles, we adopt a simple rule-based textual similarity approach using *TF-IDF*. To reduce noise from weakly related articles, we retain only the top 100 edges for each node in $K$, ensuring that the propagation graph focuses on the most relevant inter-article connections.

**Label Propagation** After constructing the label propagation graph, we apply a non-parametric label propagation algorithm to propagate observed reward signals to unobserved samples (Algorithm 2). This yields estimated rewards for all training samples, enabling effective curriculum sampling even under sparse observations.

---

**Algorithm 2** Non-Parametric Label Propagation

---

**Require:** Label propagation graph $K$, observed reward vector $\mathbf{A}$, smoothing factor $\alpha$, max iterations $T$, convergence criterion $\epsilon$
1: Normalize each row of $K$ so that $\sum_j K_{ij} = 1$
2: Initialize propagated reward $\mathbf{A}_{\text{pred}} \leftarrow \mathbf{A}$
3: **for** $t = 1$ to $T$ **do**
4:     $\mathbf{A}_{\text{new}} \leftarrow \alpha K \mathbf{A}_{\text{pred}} + (1 - \alpha) \mathbf{A}$
5:     **if** $\|\mathbf{A}_{\text{new}} - \mathbf{A}_{\text{pred}}\| < \epsilon$ **then break**
6:     **end if**
7:     $\mathbf{A}_{\text{pred}} \leftarrow \mathbf{A}_{\text{new}}$
8: **end for**
9: **return** $\mathbf{A}_{\text{pred}}$

---

**Algorithm Details** balancing the contribution between propagated information and initial observed rewards. and the maximum number of iterations is set to $T = 10$, which we found sufficient for convergence in practice. The convergence criterion is defined as $\epsilon = 10^{-4}$ These parameters are used consistently across all experiments.

## A.5 HYPERPARAMETER SENSITIVITY ANALYSIS

To assess the robustness of our method, we conduct a sensitivity analysis on two key hyperparameters: the curriculum gap threshold $\tau$ and the observation-propagation smoothing factor $\alpha$.

**Curriculum threshold $\tau$** The sensitivity analysis for the curriculum gap threshold $\tau$ is conducted within an empirically determined interval that supports meaningful curriculum progression under our KB-VQA training setup (20k InfoSeek + 20k EVQA). In preliminary diagnostics, we observe that $\tau$ values below this interval (e.g., $\tau = 0.5$) cause the model to escalate through difficulty levels too rapidly, reaching the maximum level around step 238, whereas values above it (e.g., $\tau = 0.6$) lead to stagnation, with only a single upgrade occurring at step 327. These behaviors indicate that $\tau \in (0.5, 0.6)$ forms the region in which the curriculum operates as intended, and we therefore perform the sensitivity study within this range. As shown in the left part of Figure 4, performance remains largely stable, suggesting that the method is robust to variations of $\tau$ within its effective operating regime.

**Smoothing factor $\alpha$.** For the smoothing factor $\alpha$, the sensitivity analysis is performed around the commonly adopted default configuration used in standard label-propagation implementations. Specifically, we examine $\alpha \in [0.7, 0.9]$, which forms a meaningful local neighborhood around the default choice. We evaluate the robustness of the method within this local neighborhood. As presented in the right part of Figure 4, the model maintains comparable final accuracy across all tested $\alpha$ values, indicating that the method is insensitive to moderate variations of $\alpha$ around its typical operating region.

Across both hyperparameters, the model converges to similar final accuracy within the explored intervals. Although different settings may slightly affect the rate of convergence, the eventual performance remains largely stable, suggesting that the approach is robust to hyperparameter variations and does not rely on extensive hyperparameter optimization.

## A.6 TRAINING COST COMPARISON

We compare the training cost of our method against representative baselines in Table 8. The results show that Wiki-R1 achieves competitive or lower total training costs compared to existing approaches.

Given that both Wiki-LLaVA and ReflectiVA are derived from the LLaVA-1.5 architecture and did not report their training times, we estimated their training costs based on LLaVA-1.5, using the formula: Training Time $\approx$ Baseline Time $\times$ (Data Ratio / Batch Size Ratio) $\times$ Model Size Factor.

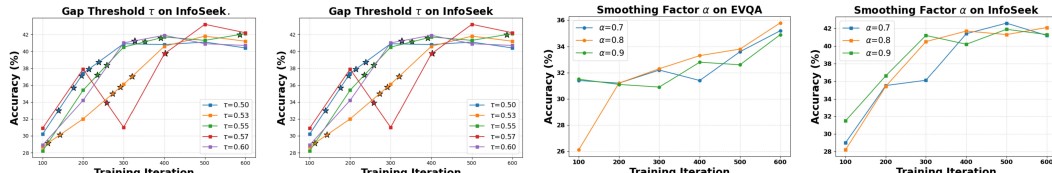

Figure 4: **Comparison across gap thresholds and smoothing factors.** We report EVQA test and InfoSeek validation performance across training iterations under different hyperparameter settings. For the left two figures, the *star* denotes an increase in curriculum difficulty during Wiki-R1 training. The chosen hyperparameter $\tau$ is 0.55 and the $\alpha$ is 0.8.

Table 8: **Training cost comparison across different methods.** FT Retrieval indicates whether the retriever is fine-tuned; FT Generation indicates whether the method fine-tunes the generation model. Training time is measured in A100 GPU hours.

| Method | FT Retrieval | FT Generation | #Training Samples EVQA | InfoSeek | Training Time |
|---|---|---|---|---|---|
| Wiki-LLaVA | × | ✓ | 916,385 | 902,509 | ~75 |
| Echosight | ✓ | × | 916,385 | 902,509 | 40 |
| ReflectiVA | × | ✓ | 2,900,000 | 2,500,000 | ~1,688 |
| Wiki-R1 (3B) | × | ✓ | 20,000 | 20,000 | 36 |
| Wiki-R1 (7B) | × | ✓ | 20,000 | 20,000 | 48 |

# B  EXPERIMENTAL STABILITY AND RELIABILITY OF RESULTS

To assess the reliability of our reported results, we conducted each experiment *three times with independent runs* under the same training settings. For each run, we recorded the performance on EVQA and InfoSeek at multiple training iterations. Figure 5 shows the performance curves for all three runs. We observe that while the *convergence speed* varies slightly across runs, the *final performance levels after convergence* are highly consistent. This indicates that our method is stable and the reported improvements are *robust to random initialization and training stochasticity*.

# C  THE USE OF LARGE LANGUAGE MODELS

In this work, large language models (LLMs) were used solely as an assistive tool for refining the writing of text authored by the researchers. Specifically, LLMs were employed to improve the readability, clarity, and conciseness of sentences drafted by the authors. All research ideas, experimental designs, analyses, and scientific claims were conceived and developed by the authors without the involvement of LLMs.

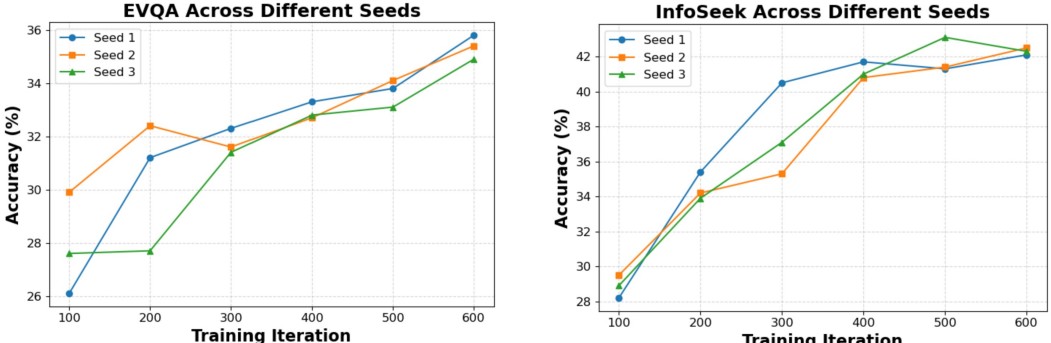

Figure 5: Performance over training iterations for three independent runs on EVQA and InfoSeek.

