# OpenReview forum: "Wiki-R1: Incentivizing Multimodal Reasoning for Knowledge-based VQA via Data and Sampling Curriculum"
_ICLR.cc/2026/Conference — ICLR 2026 Poster_

### Official Review · Reviewer_YGJ1 · 2025-10-24

**Soundness:** 1
**Presentation:** 2
**Contribution:** 2
**Rating:** 2
**Confidence:** 3

**Summary:**

The paper proposes Wiki-R1, a curriculum reinforcement learning framework based on data generation, which specifically includes three methods: (1) controlling the difficulty of samples during training, and (2) a novel sampling strategy that selects samples likely to yield significant advantages in reinforcement learning updates based on their reward values.

**Strengths:**

1) The paper’s motivation is very clear.
2) The main figure in the paper is drawn clearly.

**Weaknesses:**

1) The paper provides limited explanation of the proposed method, making it somewhat unclear.
2) In Table 2, Qwen2.5 is used, while in Table 3, Qwen2.5-VL is employed; however, the main table does not explicitly specify which LLM is used, leading to some confusion regarding the model configurations.
3) In Table 3, the inclusion of the sampling curriculum and observation propagation strategies yields inferior performance compared to using data curriculum generation alone, which undermines the effectiveness of the sampling curriculum and observation propagation strategies.
4) Table 3 does not clarify whether the SFT (Supervised Fine-Tuning) data includes ground-truth (GT) annotations, raising the possibility of an unfair comparison.
5) The method needs to be evaluated on more datasets and base models to better validate its effectiveness.

**Questions:**

See Weakness

---

> ### Author Response · Authors · 2025-11-21
>
> We sincerely thank the reviewer for the helpful comments. We address all concerns below and provide further clarification and additional evidence where needed.
>
>
> ### W1: *The paper provides limited explanation of the proposed method, making it somewhat unclear.*
>
> **We respectively disagree that a limited explanation is provided in the paper.**
> We already provide sufficient explanation of the proposed methods, as indicated by other reviewers' presentation scores.
> The pipeline of Wiki-R1-controllable curriculum data generation, curriculum sampling, and observation propagation is described step by step in Sections 3.2–3.4, further illustrated in Figure 2, and summarized again with pseudocode in Appendix A.2.
>
> We understand that the framework contains several components, and we would be happy to provide further clarification if the reviewer could indicate the specific unclear point.
>
>
> ### W2: *In Table 2, Qwen2.5 is used, while in Table 3, Qwen2.5-VL is employed; however, the main table does not explicitly specify which LLM is used, leading to some confusion regarding the model configurations.*
>
> We would like to clarify that **we consistently use Qwen2.5-VL for all our experiments.**
>
> In Table 2, the “LLM” column refers only to the language-model component of Qwen2.5-VL (i.e., the Qwen2.5 text backbone), not to a separate model used for experiments. Table 3 also uses Qwen2.5-VL in its entirety.
> To prevent ambiguity, we will revise the table caption to explicitly state that both Table 2 and Table 3 are based on Qwen2.5-VL, and that the “LLM” column in Table 2 denotes the underlying text backbone of the same model.
>
>
> ### W3: *In Table 3, the inclusion of the sampling curriculum and observation propagation strategies yields inferior performance compared to using data curriculum generation alone, which undermines the effectiveness of the sampling curriculum and observation propagation strategies.*
>
> We appreciate the reviewer’s observation.
>
> We would like to point out that **our sampling strategy brings significant data efficiency**. As shown in Figure 3 (left) of the revised paper, our sampling strategy significantly reduces the proportion of trajectories that fail to provide useful gradients (“ignored trajectories”). Furthermore, we additionally report training time: with the sampling strategy, training completes in *9 hours*, compared to *11 hours* without it—demonstrating improved *time efficiency* as well.
> Finally, we emphasize that the sampling curriculum **does not harm overall performance**. The average accuracy across EVQA and InfoSeek is:
> - Curriculum data only: 38.8
> - Curriculum data + sampling strategy: 39.1
> Thus, the sampling curriculum and observation propagation contribute meaningful efficiency gains while maintaining overall accuracy.
>
>
> ### W4: *Table 3 does not clarify whether the SFT (Supervised Fine-Tuning) data includes ground-truth (GT) annotations, raising the possibility of an unfair comparison.*
>
> Thank you for raising this important point regarding the experimental setup, which allows us to clarify a key detail.
> We confirm that the SFT (Supervised Fine-Tuning) data used in our experiments, for both our method and all baselines, did include ground-truth (GT) annotations. Therefore, the comparisons presented in Table 3 are entirely fair and conducted on an equal footing.
> To further substantiate the robustness of our method beyond this specific setting, we conducted additional ablation studies. As the reviewer will find in our response to Reviewer kWfL w3, we have included experiments under more varied conditions. These results consistently demonstrate the effectiveness of our approach and further validate our core conclusions.

---

> ### Author Response · Authors · 2025-11-21
>
> ### W5: *The method needs to be evaluated on more datasets and base models to better validate its effectiveness.*
>
> **Regarding More Datasets**
>
> We have conducted experiments on the most representative KB-VQA benchmarks, including Infoseek and EVQA. To further validate the generality of our approach, we have additionally evaluated our method on ViQuAE, which demonstrates consistent effectiveness across datasets.
>
> Specifically, we conduct **zero-shot transfer** evaluation on the ViQuAE test set following the official protocol. We compute *F1* and *Exact Match* scores using the official evaluation code, and adopt the 1.4M Wikipedia article knowledge base provided by ViQuAE. For comparison, we report the baseline RC results from Table 4 of the ViQuAE [6] paper and the MLLM-based results from Table 9 of ReflectiVA [7]. The detailed results are shown in the table below:
>
> | Model| F1 | Exact Match |
> |-------------|----|-------------|
> | **RC Baseline** |||
> | RC Zero-shot              | 20.96 | 18.06 |
> | RC Few-shot               | 25.43 | 22.07 |
> | RC Few-shot (semi-oracle) | 44.10 | 40.32 |
> | RC Few-shot (full-oracle) | 63.17 | 57.55 |
> | **MLLM Baseline** |||
> | LLaVA-v1.5                | 15.1  | 26.6  |
> | Wiki-LLaVA (E-VQA)        | 10.5  | 16.7  |
> | Wiki-LLaVA (InfoSeek)     | 12.7  | 21.8  |
> | ReflectiVA                | 23.2  | 38.1  |
> | **Ours** |||
> | Wiki-R1 3B                | 53.8  | 48.6  |
> | Wiki-R1 7B                | **55.6**  | **50.3**  |
>
>
> From the results, we observe that Wiki-R1 significantly outperforms all baselines, and even surpasses the RC (semi-oracle) setting. This demonstrates the strong generalization ability of our method under zero-shot cross-dataset transfer.
>
> **Regarding More Base Models**
>
> We respectfully argue that the base models used in our experiments are already sufficient to support the claims of this paper.
> Specifically, we evaluate our framework on two variants of the recent state-of-the-art Qwen2.5-VL model family (3B and 7B). These models are not only strong representatives of current multimodal LLMs, but are also the standard backbones adopted in many very recent multimodal RL works[1,2,3,4].
> In addition, there is a substantial performance gap between earlier base models (e.g., LLaVA 1.5) and Qwen2.5-VL. Therefore, we do not consider these earlier models as suitable base models for comparison.
>
> Overall, our choice of datasets and base models sufficiently supports the claims of our work, and the additional experiments on ViQuAE further reinforce the generality and effectiveness of our method.
>
> Reference:
>
> [1] VLM-R1: A Stable and Generalizable R1-style Large Vision-Language Model. Arxiv 25.4 (citation 245)
>
> [2] Perception R1: Pioneering Perception Policy with Reinforcement Learning. NeurIPS 25
>
> [3] Open Vision Reasoner: Transferring Linguistic Cognitive Behavior for Visual Reasoning. NeurIPS 25
>
> [4] NoisyGRPO: Incentivizing Multimodal CoT Reasoning via Noise Injection and Bayesian Estimation. NeurIPS 25
>
> [5] Wiki-LLaVA: Hierarchical Retrieval-Augmented Generation for Multimodal LLMs. CVPR24 Workshop
>
> [6]  ViQuAE, a Dataset for Knowledge-based Visual Question Answering about Named Entities.
>
> [7] Augmenting Multimodal LLMs with Self-Reflective Tokens for Knowledge-based Visual Question Answering. CVPR24

---

> ### Comment · Reviewer_YGJ1 · 2025-11-25
> **Official Comment by Reviewer YGJ1**
>
> Thanks a lot for the authors’ thoughtful and patient responses.
>
> **W1**: *The paper provides limited explanation of the proposed method, making it somewhat unclear.*
>
> More intuitively, the "Weaknesses" section primarily outlines the motivation for the methodological design. For instance, in Controllable Data Generation, the easiest level trains the model to answer directly from a clean document, while intermediate levels progressively challenge it to locate the ground-truth answer in increasingly noisy contexts. However, the hardest level is puzzling, as it appears to train the model to handle nosiy knowledge ( RAG’s assumption that the retrieved context contains the correct answer).
>
> Furthermore, the sampling schedule, which targets a training accuracy near 0.5, seems only weakly tied to difficulty.  A more natural approach would be to advance only after the model has sufficiently adapted to the current level.
>
>
> **W3**: *In Table 3, the inclusion of the sampling curriculum and observation propagation strategies yields inferior performance compared to using data curriculum generation alone, which undermines the effectiveness of the sampling curriculum and observation propagation strategies.*
>
> The authors’ response on this point remains unclear to me. Given that the sampling schedule is intended to control difficulty, its effectiveness should be measured by accuracy rather than training time. Additionally, in Table 3, the result for “Curriculum data + sampling strategy” is 36.1, not 39.1. It is recommended that this aspect be prioritized for improvement in future work.
>
> **W4**: *Table 3 does not clarify whether the SFT (Supervised Fine-Tuning) data includes ground-truth (GT) annotations, raising the possibility of an unfair comparison.*
>
> Let me double-check: after SFT, is the model evaluated using the base model, or with a given retrieved document? Since SFT performance should not be substantially lower than that of RL.
>
> **W5**: *The method needs to be evaluated on more datasets and base models to better validate its effectiveness.*
>
> Thank you for the additional experiments, I have no further comments on this weakness.

---

> ### Author Response · Authors · 2025-11-26
> **Response to Reviewer Comments - Second Round**
>
> We thank the reviewer for their time and the opportunity to provide further clarifications.
>
> # W1
> ### (1) Motivation for including the hardest difficulty level (training on "noisy knowledge").
>
>  Thank you for correctly understanding the motivation behind the earlier difficulty levels.
> We would like to briefly clarify that **we do not assume that the retrieved context necessarily contains the correct answer.** Such an assumption is unrealistic since real-world retrieval is often incomplete or noisy, as noted in the introduction section of our main paper.
>
> Therefore, using actual noisy retrieval outputs as the highest-difficulty stage is a natural and necessary part of the curriculum, aligning the training distribution with the model’s true inference-time conditions.
>
> ### (2) Motivation for targeting a training accuracy near 0.5 in the sampling schedule.
>
>  As described in Section 3.4, our sampling schedule is inspired by prior work in curriculum learning for reinforcement finetuning (Efficient Reinforcement Finetuning via Adaptive Curriculum Learning). This work shows that **samples with a model accuracy near 0.5 provide the strongest gradient signal and thus offer the highest learning potential for RL-based optimization.**
>
> Our goal is therefore not to interpret accuracy as a direct measure of semantic difficulty, but rather to identify samples that maximize learning efficiency. The sampling strategy is designed to effectively search for and prioritize these informative samples, which improves training efficiency.
>
> We agree that other alternative methods for sampling scheduling remain an interesting future direction. However, empirical evidences show that such a sampling schedule is effective under this specific problem.
>
> # W3
> ### (1) Clarification regarding the results reported in Table 3
>
> The comment is mistakenly referring to “Curriculum data + sampling curriculum”, which is not the complete sampling strategy
>
> The complete sampling strategy includes **both the sampling curriculum and observation propagation**, as also noted in your initial comments. The row corresponding to the full strategy is therefore “Curriculum data + sampling curriculum + observation propagation,” which achieves an average accuracy of 39.1. The average accuracy is improved compared with the method without our sampling strategy, which is an average accuracy of 38.8 in row “Curriculum data only”.
>
> ### (2) Evaluation of the sampling schedule's effectiveness.
>
> We would like to clarify that **even if the sampling schedule seems to be related to difficulty control, it does not necessarily follow that “its effectiveness should therefore be evaluated by only accuracy.”**
>
> The sampling strategy is primarily designed to improve training efficiency, which is a critical challenge in RL, and we have provided a detailed explanation in the main paper.
>
> In the ablation study section, we already measure the contribution to both accuracy and efficiency of the sampling strategy. As shown in Table 3 and Figure 3, **the full sampling strategy improves both data and time efficiency and improves overall accuracy**, aligning with its intended purpose.
>
> # W4
>
> ### (1) SFT configuration
>
> We appreciate the reviewer’s question and provide the detailed SFT configuration as follows:
>
> - What retrieval configuration was used for the SFT baseline?
>
> Both RL and SFT are provided with retrieval documents, which are derived from the same retrieval models (EVA-CLIP-8B + ColBERT).
>
> - Whether the SFT data includes ground-truth (GT) annotations?
>
> During training, the GT answers to questions are provided for both RL and SFT.
>
> ### (2) Why RL outperforms SFT
>
> As we have demonstrated in the introduction section of our main paper, KB-VQA is a challenging task that demands robust reasoning ability. Previous works have shown that RL methods possess superior and more stable reasoning capabilities, making them well-suited for this task. Therefore, it's not surprising that RL outperforms SFT on this task.
>
> # W5
> We appreciate the reviewer’s feedback. We are glad that the additional experiments addressed this concern, and we thank the reviewer for the positive acknowledgment.

---

> > ### Comment · Reviewer_YGJ1 · 2025-11-28
> > **Official Comment by Reviewer YGJ1 - Second Round**
> >
> > Thank you for your patient response. Below are my remaining concerns:
> >
> > **W1**: *Motivation for including the hardest difficulty level (training on "noisy knowledge").*
> >
> > Yes, the retrieved context does not necessarily contain the correct answer; however, the goal of RAG is to enable the model to correctly identify and effectively use the retrieved information when it does contain the ground-truth answer. Therefore, I remain puzzled by this motivation.
> >
> > **W3**: *Clarification regarding the results reported in Table 3*
> >
> > If the full sampling strategy is not used, could you please clarify what the ablation study for "Curriculum data + sampling curriculum" actually entails? I’m still somewhat confused.
> >
> > **W4**: *Why RL outperforms SFT*
> >
> > Thank you for your patient response, I have no confusion regarding this issue.

---

> ### Author Response · Authors · 2025-11-28
> **Response to Reviewer Comments - Third Round**
>
> We thank the reviewers for their time and valuable comments. Below, we provide a point-by-point response to the comments.
>
> # W1
>
> We would like to clarify that our motivation explicitly addresses the realistic scenario where retrieval contains inherent noise. The RAG framework can be divided into retrieval and generation. Our work aims to improve the model's generation ability under such noisy retrieval, which is parallel to another trend of works focusing on improving retrieval quality. Specifically, in the last difficulty level, the model is trained to adapt to real-world retrieval noise, improving its robustness and aligning the training distribution with the target inference distribution.
>
> # W3
>
> As described in Sec. 3.4, the sampling strategy consists of two operations: curriculum sampling and observation propagation.
> - Curriculum sampling: Training accuracy of observed samples is recorded, and during training, samples with a desired training accuracy (0.5 in our case) are prioritized.
> - Observation propagation: Because relying solely on observed training samples is inefficient, we propagate training observations to unobserved samples to improve sampling.
>
> In the ablation study:
>
> - "Curriculum data + sampling curriculum" uses only curriculum sampling (sampling based on observed training accuracy) without observation propagation.
> - "Curriculum data + sampling curriculum + observation propagation" uses the full sampling strategy (curriculum sampling enhanced by observation propagation).
>
> # W4
>
> Thank you for the confirmation. We appreciate your careful reading and are glad the issue is now clear.
>
> *We hope this response helps the reviewer better understand the contributions of our work.*

---

### Official Review · Reviewer_zx7D · 2025-10-26

**Soundness:** 3
**Presentation:** 3
**Contribution:** 3
**Rating:** 6
**Confidence:** 4

**Summary:**

This paper proposes Wiki-R1, a data-generation-based curriculum reinforcement learning framework designed to enhance multimodal large language models (MLLMs) on Knowledge-Based Visual Question Answering (KB-VQA) tasks.
The key idea is to systematically bridge the distribution gap between pretraining data and the KB-VQA domain by Controllable Curriculum Data Generation and Curriculum Sampling with Observation Propagation.
Through these mechanisms, the model learns progressively harder reasoning examples, mitigating sparse rewards during RL fine-tuning.
Experiments on Encyclopedic-VQA and InfoSeek show new state-of-the-art performance with strong generalization to unseen questions.

**Strengths:**

1. The framework introduces an elegant combination of data-level and sampling-level curricula. The idea of controlling retrieval difficulty rather than merely selecting data is innovative and well-motivated by the sparse reward challenge in KB-VQA.
2. The approach yields notable accuracy gains with only ~40k training samples — far less than prior methods requiring millions — highlighting efficiency and scalability.

**Weaknesses:**

1. While the combination of controllable retrieval and curriculum sampling is well-engineered, the theoretical novelty may be seen as incremental over prior curriculum RL works. The core mechanism (progressively harder data and adaptive sampling) is conceptually similar.
2. The performance gains are mostly demonstrated on EVQA and InfoSeek. It’s unclear whether the framework generalizes to other KB-VQA settings (e.g., OK-VQA) or to different retrieval model architectures.
3. The literature review seems to focus on works on EVQA/InfoSeek. A large collection of works in other KB-VQA datasets (e.g. OK-VQA) are missed in the discussion.
4. The proposed model outperforms many existing works, however, it remains unclear to me whether the gain is from RL training since many existing works are training-free. The authors should make it clear in the table (whether the approach is training-free) and compare the approach with training methods.

**Questions:**

The paper defines an upgrade threshold τ = 0.55 for moving to the next difficulty level. How sensitive is performance to this threshold? Would too-rapid or too-slow progression harm training stability?

---

> ### Author Response · Authors · 2025-11-21
>
> We are grateful for the reviewer’s thoughtful evaluation and valuable input. We have carefully examined the raised points, and our responses and clarifications are provided as follows.
>
> ### W1: *While the combination of controllable retrieval and curriculum sampling is well-engineered, the theoretical novelty may be seen as incremental over prior curriculum RL works. The core mechanism (progressively harder data and adaptive sampling) is conceptually similar.*
>
> To conceptually summarize, the main novelty of our work is the introduction of **an efficient difficulty-estimation mechanism for curriculum learning**, which relies only on easily accessible priors rather than manual or model-derived annotations. In other words, it can be viewed as  **performing difficulty estimation in an unsupervised manner**, addressing a limitation of prior curriculum RL methods [1,2,3].
> Specifically, our contributions consist of two components: (1) curriculum data generation and (2) a label-propagation–based sampling strategy.
> - **Curriculum data generation**: We demonstrate that simple priors—such as the number of retrieved candidates or whether the ground-truth evidence is guaranteed to be present—are sufficient to construct samples with automatically known difficulty levels. By contrast, prior works typically require annotated difficulty labels or rely on external models’ inference scores to approximate difficulty.
> - **Sampling strategy**: We leverage the relationships among samples as priors, allowing the model’s difficulty observations to propagate in a fully unsupervised manner. Previous curriculum RL approaches generally lack such a principled mechanism to expand sparse difficulty signals.
> In summary, while the high-level intuition of progressive curriculum RL is shared, our method provides a simple, practical, and self-contained solution for difficulty estimation and sampling, effectively addressing a previously unmet need in the literature.
>
> Reference:
>
> [1] Efficient Reinforcement Finetuning via Adaptive Curriculum Learning.
>
> [2] VL-Cogito: Progressive Curriculum Reinforcement Learning for Advanced Multimodal Reasoning.
>
> [3] A Curriculum Learning Approach to Reinforcement Learning: Leveraging RAG for Multimodal Question Answering.

---

> ### Author Response · Authors · 2025-11-21
>
> ### W2: *The performance gains are mostly demonstrated on EVQA and InfoSeek. It’s unclear whether the framework generalizes to other KB-VQA settings (e.g., OK-VQA) or to different retrieval model architectures.*
>
> Thank you for the helpful suggestion.
>
> **Generalization over other KB-VQA**
> We demonstrate the generalization of Wiki-R1 on the additional KB-VQA benchmark, ViQuAE[1].
> As noted in the InfoSeek[2] paper (Sec. 2 *"The Need for a New Visual Information-seeking Benchmark"*), OK-VQA mainly tests common-sense knowledge rather than true information-seeking ability, so it is less suitable for our setting. So we chose the more recent and challenging ViQuAE as a better testbed.
> Specifically, we conduct **zero-shot transfer** evaluation on the ViQuAE test set following the official protocol. We compute *F1* and *Exact Match* scores using the official evaluation code, and adopt the 1.4M Wikipedia article knowledge base provided by ViQuAE. For comparison, we report the baseline RC results from Table 4 of the ViQuAE [1] paper and the MLLM-based results from Table 9 of ReflectiVA [3]. The detailed results are shown in the table below:
>
> | Model| F1 | Exact Match |
> |-------------|----|-------------|
> | **RC Baseline** |||
> | RC Zero-shot              | 20.96 | 18.06 |
> | RC Few-shot               | 25.43 | 22.07 |
> | RC Few-shot (semi-oracle) | 44.10 | 40.32 |
> | RC Few-shot (full-oracle) | 63.17 | 57.55 |
> | **MLLM Baseline** |||
> | LLaVA-v1.5                | 15.1  | 26.6  |
> | Wiki-LLaVA (E-VQA)        | 10.5  | 16.7  |
> | Wiki-LLaVA (InfoSeek)     | 12.7  | 21.8  |
> | ReflectiVA                | 23.2  | 38.1  |
> | **Ours** |||
> | Wiki-R1 3B                | 53.8  | 48.6  |
> | Wiki-R1 7B                | **55.6**  | **50.3**  |
>
>
> From the results, we observe that Wiki-R1 significantly outperforms all baselines, and even surpasses the RC (semi-oracle) setting. This demonstrates the strong generalization ability of our method under zero-shot cross-dataset transfer.
>
> **Generalization over retrieval model**
> To further evaluate whether our framework generalizes to different retrieval model architectures, we conducted an additional study under a *purely visual retrieval setting* using CLIP I-I similarity.
>
> Specifically, we trained the policy using our Wiki-R1 retrieval module—which jointly leverages *both textual and visual* retrieval—and then **directly evaluated the resulting model under a different retrieval setting (CLIP I-I) without any retraining or adaptation**. The detailed retrieval recall of both systems is reported in the main paper Appendix Table 4.
> For a fair comparison, we use DAPO trained under the same settings as the baseline. The performance comparison is reported in the table below.
>
> | Method   | EVQA Single-hop | EVQA All | InfoSeek  Unseen-Q | InfoSeek Unseen-E | InfoSeek All | Avg  |
> |----------|----------------|----------|---------------|---------------|--------------|------|
> | DAPO     | 28.1           | 27.0     | 42.4          | 38.0          | 40.2         | 33.6 |
> | Wiki-R1  | 31.3           | 30.2     | 45.2          | 39.2          | 41.6         | 35.9 |
>
>
> From the table, we observe that our framework consistently outperforms the DAPO baseline across different retrieval settings. Under the I2I setting, both Wiki-R1 and the baseline exhibit performance drops. We believe this is a common issue in RAG-based KB-VQA, where weaker retrieval naturally leads to weaker VQA performance. Nevertheless, when using the stronger Wiki-R1 retrieval system, our model achieves a larger improvement over the baseline (6.8% vs. 7.4%), demonstrating that our framework benefits more from better retrieval quality.
> These demonstrate that *our method remains effective and robust when applied to varying retrieval architectures.*
>
>
> Reference:
>
> [1] ViQuAE, a Dataset for Knowledge-based Visual Question Answering about Named Entities.
>
> [2] Can Pre-trained Vision and Language Models Answer Visual Information-Seeking Questions?
>
> [3] Augmenting Multimodal LLMs with Self-Reflective Tokens for Knowledge-based Visual Question Answering.

---

> > ### Author Response · Authors · 2025-11-21
> >
> > ### W3: *The literature review seems to focus on works on EVQA/InfoSeek. A large collection of works in other KB-VQA datasets (e.g. OK-VQA) are missed in the discussion.*
> >
> > Thank you for this insightful comment. We agree that discussing the broader landscape of KB-VQA datasets is valuable for providing context.
> > We address this by adding a discussion of OK-VQA and other early KB-VQA benchmarks in the revised paper Section 2.1 (highlighted in blue).
> > The primary reason our work focuses on the InfoSeek and EVQA benchmarks, rather than OK-VQA, is that OK-VQA's characteristics do not fully align with the core research problem we are investigating, as noted in the InfoSeek[1] paper (Sec. 2 *"The Need for a New Visual Information-seeking Benchmark"*). Specifically:
> > 1. Lack of Entity Annotation: OK-VQA does not provide explicit annotations for the entities in its questions, which play significant role in our method.
> > 2. Different Problem Nature: The questions in OK-VQA are often more reliant on common sense and are answerable by a knowledgeable adult, whereas InfoSeek and EVQA specifically require the retrieval of specific, factual knowledge, which is the central challenge our paper aims to address. This difference in difficulty and focus is reflected in the performance gaps reported in prior work between models on OK-VQA versus these more demanding benchmarks.
> >
> >
> > Reference:
> >
> > [1] Can Pre-trained Vision and Language Models Answer Visual Information-Seeking Questions?
> >
> >
> >
> > ### W4: *The proposed model outperforms many existing works, however, it remains unclear to me whether the gain is from RL training since many existing works are training-free. The authors should make it clear in the table (whether the approach is training-free) and compare the approach with training methods.*
> >
> > Thank you for raising this important point regarding the source of our performance gains.
> > We would like to clarify that all baselines we compared against are trained approaches. The performance gain is therefore a direct comparison between different training paradigms, and not a comparison against training-free methods.
> >
> > ### Q1: *The paper defines an upgrade threshold τ = 0.55 for moving to the next difficulty level. How sensitive is performance to this threshold? Would too-rapid or too-slow progression harm training stability?*
> >
> > We thank the reviewer for pointing out this concern.
> >
> > To verify the robustness of our method, we conducted a sensitivity analysis on the two key hyperparameters in our framework—the curriculum gap threshold τ and the observation propagation smoothing factor α.
> > As shown in the revised paper Figure 4 in the revised paper, the model achieves consistent final performance across a wide and reasonable range of τ and α values. While different hyperparameter choices may slightly affect the convergence speed, the final accuracy remains largely stable.
> > These findings demonstrate that our method is insensitive to the exact hyperparameter choices and remains robust without requiring extensive hyperparameter tuning, supporting its practicality in real-world applications.

---

> ### Comment · Reviewer_zx7D · 2025-11-23
>
> Thanks for the response.
>
> I agree that conventional KB-VQA datasets like OK-VQA have certain limitations. However, it is still necessary to discuss earlier systems that were originally evaluated on OK-VQA. To my knowledge, many systems that perform well on OK-VQA also generalize effectively to E-VQA, since they share the same underlying principles. I believe the authors should at least cite some representative OK-VQA systems. Although these systems were not evaluated on newer datasets, they are by no means weak or irrelevant.

---

> > ### Author Response · Authors · 2025-11-24
> >
> > Thank you for the helpful suggestion. We agree that earlier systems evaluated on OK-VQA remain relevant, and we have now added a discussion of representative OK-VQA methods in the revised manuscript (Section 2.1, highlighted in blue). We appreciate the pointer, and if you believe we have overlooked any important prior work, we would be grateful if you could let us know so that we can further improve the paper.

---

> ### Comment · Reviewer_zx7D · 2025-11-27
>
> The extended discussion looks fine. As most of my concerns have been addressed, I will keep the positive score.

---

### Official Review · Reviewer_Wc3T · 2025-10-27

**Soundness:** 3
**Presentation:** 3
**Contribution:** 3
**Rating:** 4
**Confidence:** 4

**Summary:**

This paper identifies the distribution gap with KB-VQA data which leads to sparse rewards in RL approaches and proposes Wiki-R1, a curriculum RL framework with two core components: Controllable Curriculum Data Generation and Curriculum Sampling with Observation Propagation. Experiments on Encyclopedic VQA (EVQA) and InfoSeek benchmarks show state-of-the-art (SOTA) results.

**Strengths:**

1. Targeted Problem: Addresses a critical pain point of distribution gap and sparse rewards in RL-based KB-VQA.
2. Generalization: Excels on unseen question splits e.g., InfoSeek Unseen-Q:47.8% vs. prior 40.4, indicating robust reasoning.
3. Component Validity: Ablations clearly show that data curriculum improves DAPO performance, and propagation is necessary for sampling curriculum to work).

**Weaknesses:**

1. Limited Retrieval Control: The data generation relies on adjusting retrieval noise (number of candidates, ground-truth inclusion) but does not fully control the type of noise (e.g., irrelevant vs. slightly relevant candidates).
2. Hyperparameter Transparency: No sensitivity analysis for key hyperparameters (e.g., curriculum gap threshold τ, observation propagation smoothing factor α).
3. RL Algorithm Scope: Only uses DAPO as the base RL algorithm—no comparison with other RL methods (e.g., PPO, GRPO) to validate generalizability.

**Questions:**

See Weaknesses.

---

> ### Author Response · Authors · 2025-11-21
>
> We appreciate the reviewer’s insightful comments and suggestions. We provide detailed responses to each concern below.
>
> ### W1: *Limited Retrieval Control: The data generation relies on adjusting retrieval noise (number of candidates, ground-truth inclusion) but does not fully control the type of noise (e.g., irrelevant vs. slightly relevant candidates).*
>
> We would like to clarify that our goal in adjusting retrieval noise is to create a progressively more challenging distribution that supports curriculum learning and helps the model receive more stable gradients during training. **Our partial retrieval control is sufficient to achieve the intended effect.** Details are provided below.
> - Our partial control over retrieval noise has been shown to improve training stability. Wiki-R1 reduces the number of samples without useful gradient compared with the baseline, thus providing stable training signals as shown in Figure 3 (left).
> - Moreover, the evaluation curves over training iterations further indicate the stabilizing effect of our curriculum strategy. Specifically, as shown in Figure 3 (right), while our method begins with lower performance, it improves steadily as the difficulty level increases, whereas the baseline starts higher but later plateaus and even degrades.
>
> We agree that more fine-grained control over noise types is an interesting direction. While such enhancements may further improve performance, **we view them as promising future work rather than a limitation of our current approach**, given that the existing strategy already effectively fulfills its role within our curriculum-learning framework.
>
>
>
> ### W2: *Hyperparameter Transparency: No sensitivity analysis for key hyperparameters (e.g., curriculum gap threshold τ, observation propagation smoothing factor α).*
>
> We thank the reviewer for pointing out this concern.
>
> To verify the robustness of our method, we conducted a sensitivity analysis on the two key hyperparameters in our framework—the curriculum gap threshold \tau and the observation propagation smoothing factor α.
> As shown in the revised paper Figure 4 (highlighted in blue) in the revised paper, the model achieves consistent final performance across a wide and reasonable range of τ and α values. While different hyperparameter choices may slightly affect the convergence speed, the final accuracy remains largely stable.
> These findings demonstrate that our method is **insensitive to the exact hyperparameter choices** and remains robust without requiring extensive hyperparameter tuning, supporting its practicality in real-world applications.
>
>
> ### W3: *RL Algorithm Scope: Only uses DAPO as the base RL algorithm—no comparison with other RL methods (e.g., PPO, GRPO) to validate generalizability.*
>
> We thank the reviewer for the helpful suggestion.
>
>  To evaluate whether our framework generalizes beyond DAPO, we additionally applied it to GRPO, a widely used value-free RL algorithm. As shown in the table below, Wiki-R1 GRPO consistently outperforms the vanilla GRPO baseline, demonstrating that our framework is not tied to a specific RL algorithm.
>
>
> | Method        | EVQA Single-hop | EVQA All |InfoSeek Unseen-Q  | InfoSeek Unseen-E | InfoSeek All | Average |
> |---------------|----------|------------------|----------|----------------|----------------|---------------|
> | GRPO          | 30.8     | 27.0             | 37.3     | 24.4           | 28.7           | 27.8          |
> | Wiki-R1 GRPO  | 34.2     | 30.5             | 38.2     | 24.6           | 29.2           | 29.7          |
>
>
> We would also like to clarify why the GRPO is noticeably weaker than DAPO. This difference is expected and aligns with the motivation of our work: GRPO lacks the dynamic filtering mechanism in DAPO, and therefore suffers more severely from distribution mismatch and sparse-reward issues, which hinder stable learning when the training and target distributions differ substantially.
> For this reason, DAPO is a more suitable primary baseline for our setting, while GRPO is included to further verify the generalizability of our framework.
>
> Regarding PPO, we did not include it in our experiments because PPO requires training an additional value model, which introduces extra sources of variance and confounding factors. Since our goal is to isolate the effect of the curriculum RL framework, we focus on value-free RL algorithms DAPO that avoid these confounding effects.

---

> > ### Comment · Reviewer_Wc3T · 2025-11-27
> > **Response to Authors**
> >
> > Thank you for your rebuttal. Your reply partly addressed my concern.
> >
> > However, the experimental setup for W2 is insufficient. Standard practice for hyperparameter sensitivity involves covering a practically relevant range and sampling multiple points. Yet, the experiments in the manuscript only explore a narrow interval ($\tau$ 0.53-0.57, step 0.02; $\alpha$ 0.7-0.9, step 0.1). The authors provide no justification for assuming that optimal values lie within this constrained interval. The paper does not explain why the sensitivity analysis was conducted solely within this extremely narrow range, which raises concerns about the sufficiency of this experiment. Consequently, I will maintain my score.

---

> > > ### Author Response · Authors · 2025-12-03
> > > **Response to Reviewer Comments - Second Round**
> > >
> > > We appreciate the reviewer’s continued feedback. We provide justification for the choice of experimental setup for sensitivity analysis.
> > >
> > > **(1) Curriculum threshold τ.**
> > >
> > > We clarify the rationale behind the choice of τ. We seek an operating range of τ where curriculum progression is neither overly aggressive nor stalled, since either extreme would undermine progressive learning. We conduct analysis on our KB-VQA training dataset consisting of 20k InfoSeek and 20k EVQA training data.
> > >
> > > Empirically, under this experimental setting, τ ∈ [0.5, 0.6] satisfies this criterion. As shown in the revised Appendix A.5:
> > >
> > > - When τ = 0.5, the curriculum escalates too rapidly, reaching its maximum difficulty as early as step 238.
> > > - When τ = 0.6, the model upgrades only once and never satisfies the promotion condition thereafter.
> > >
> > > Thus, τ outside this interval either induces overly fast advancement or prevents meaningful progression, making [0.5, 0.6] a reasonable working band for our setting. The same protocol can also be applied to select the corresponding reasonable range for other datasets.
> > >
> > > Within this range, we perform a sensitivity analysis to assess robustness. The results in Appendix A.5 show that performance remains stable across τ ∈ [0.5, 0.6], indicating that our framework is not sensitive to the precise choice of τ within this band.
> > >
> > > **(2) Smoothing factor α.**
> > >
> > > We appreciate the reviewer’s request for a broader ablation study on the smoothing factor α and clarify the rationale of our sensitivity analysis design below.
> > >
> > > **Choice of α.** We follow common practice in label-propagation implementations (e.g., scikit-learn) and adopt the widely used default value as our reference configuration. This provides a practically grounded starting point without extensive tuning.
> > >
> > > **Local stability validation.** Based on this reference, we allocate our available computational budget to examine robustness in its vicinity. Specifically, we evaluate α within the interval [0.7, 0.9], which forms a meaningful local neighborhood around the default choice. As shown in the revised Appendix A.5, the results exhibit consistent performance, suggesting that the method is locally stable and insensitive to moderate perturbations of α.
> > >
> > > **Extension to a broader range.** While this analysis validates stability within the commonly used regime, we agree that exploring a wider span of α would strengthen generality. We will expand this investigation to cover a broader range in the final version.

---

> ### Author Response · Authors · 2025-11-26
> **Reminder for rebuttal**
>
> Dear Reviewer Wc3T,
>
> Thank you once again for your valuable feedback during the first round of review. As the discussion phase is scheduled to conclude on December 3, we would like to kindly check whether all the issues have been properly addressed from your perspective. We also warmly welcome any further comments or suggestions you might have.
>
> If you feel that all concerns have been resolved satisfactorily, we would be grateful if you could consider raising the scores.
> Best regards, Authors

---

### Official Review · Reviewer_kWfL · 2025-10-31

**Soundness:** 3
**Presentation:** 3
**Contribution:** 3
**Rating:** 6
**Confidence:** 4

**Summary:**

This paper addresses the task of enhancing multimodal large language models (MLLMs) for Knowledge-Based Visual Question Answering (KB-VQA), where models need to integrate visual and textual information to answer questions. The authors propose Wiki-R1, a reinforcement learning framework with two main contributions. First, Curriculum Data Generation manipulates the retriever to create training samples at different difficulty levels, enabling the model to learn progressively from easy (accurate retrieval) to hard (noisy retrieval) examples. Second, Curriculum Sampling with Observation Propagation implements a sampling strategy that prioritizes the most informative samples (those with approximately 50% accuracy) and propagates difficulty estimates to unseen samples. These components work together with RL optimization to provide denser reward signals and improve reasoning capabilities in KB-VQA scenarios. The method is evaluated on two KB-VQA benchmarks.

**Strengths:**

- The paper clearly identifies why RL fails in KB-VQA (retrieval noise leads to sparse rewards) and proposes a reasonable data-centric solution to increase reward density and improve downstream reasoning performance.
- The core idea is elegant: generating progressive difficulty data through controllable parameters and selecting the most valuable samples using accuracy-based Gaussian sampling to balance between "already learned" and "not yet learned" examples. I particularly appreciate the Observation Propagation mechanism for finding similar samples to stabilize training.
- The method shows consistent performance gains and potential better data efficiency compared to baselines.

**Weaknesses:**

- Some key parameters are not specified, including the exact definition of the reward function in the RL objective, and the observation propagation parameters.
- The reliance on TF-IDF similarity may restrict the method to surface-level lexical matching, potentially missing semantically similar samples that use different vocabulary.
- The paper lacks details needed to assess the true benefit of RL. Specifically: What retrieval configuration was used for the SFT baseline? Does it use the same samples as the RL experiments? Was the SFT training data also curriculum-generated? Without a fair comparison where both SFT and RL use identical curriculum data, it's difficult to isolate whether the gains come from the curriculum design itself or from the RL optimization.
- Results are reported as single values without standard deviations or significance tests, making it difficult to assess the reliability of the comparisons.
- Table 5's comparison may be misleading because it only lists training sample counts without reporting actual training time, GPU costs, or convergence speed. While Wiki-R1 achieves better results with less data, it employs computationally intensive RL training (multiple rollouts per sample, dynamic sampling, observation propagation), whereas baselines use simpler SFT. The paper reports Wiki-R1's training time but provides no training time data for baselines, making it hard to determine which method has lower total computational cost.

**Questions:**

What exact data configuration was used for the SFT baseline in Table 3? Does it use the same entity-balanced samples as the RL experiments?

---

> ### Author Response · Authors · 2025-11-21
>
> We thank the reviewer for the constructive feedback. Below, we address the reviewer’s concerns point by point and clarify the corresponding aspects of the paper.
>
> ### W1: *Some key parameters are not specified, including the exact definition of the reward function in the RL objective, and the observation propagation parameters.*
>
> Thank you for pointing out the missing details. We clarify the following:
> - **Reward function**: We use a rule-based reward: the reward is 1 if the predicted answer exactly matches the ground-truth answer, and 0 otherwise.
> - **Observation propagation parameters**: All parameters used in Algorithm 2 of the main paper are:
>   - smoothing factor: alpha = 0.9
>   - maximum iterations: T = 10
>   - convergence criterion: epsilon=1e-4
>
> We would like to clarify that we do not tune these parameters; instead, we adopt standard and commonly used default values. We also verified that varying these parameters within a reasonable range does not materially affect the results, indicating that our method is not sensitive to them. Specifically, we conducted an ablation on the smoothing factor, and the results are provided in the revised paper (Figure 4 and Appendix Section A.5, highlighted in blue). The results show that our method is robust to the Label Propagation parameter choices. We will add the above details to the final version of the paper.
>
> ### W2: *The reliance on TF-IDF similarity may restrict the method to surface-level lexical matching, potentially missing semantically similar samples that use different vocabulary.*
>
> We sincerely thank the reviewer for this helpful observation.
>
> We would like to clarify that our focus is on the label propagation framework, and the chosen similarity function is intended as a simple instantiation. To better understand the effect of the similarity measure used to build the label graph, we additionally experimented with a text embedding model (Sentence Transformer). The results are shown in the table below.
>
> | Method        |  Single-hop EVQA | EVQA All  | InfoSeek Unseen-Q | InfoSeek Unseen-E | InfoSeek All | Avg  |
> |---------------|----------|----------------|---------------|---------------|--------------|------|
> | TF-IDF        | 40.4     | 35.9           | 46.0          | 40.3          | 42.2         | 39.1 |
> | Sentence Transformer | 41.1     | 36.3           | 46.2          | 41.3          | 43.0         | 40.7 |
>
> These results show that our label propagation framework is compatible with various semantically-aware similarity measures and can potentially benefit from them. A detailed study of different similarity functions, however, is beyond the scope of this work and is left for future research.
>
>
> ### W3: *The paper lacks details needed to assess the true benefit of RL. Specifically: What retrieval configuration was used for the SFT baseline? Does it use the same samples as the RL experiments? Was the SFT training data also curriculum-generated? Without a fair comparison where both SFT and RL use identical curriculum data, it's difficult to isolate whether the gains come from the curriculum design itself or from the RL optimization.*
>
> We appreciate the reviewer’s question and provide the detailed answers as follows:
>
> - *What retrieval configuration was used for the SFT baseline? *
>
>  The SFT baseline uses the same retrieval models as the RL experiments (EVA-CLIP-8B + ColBERT).
>
> - *Does it use the same samples as the RL experiments?*
>
>  SFT is trained on the same dataset as in the RL setting (20k EVQA + 20k Infoseek).
>
>  It also uses the same entity-balanced samples as the RL experiments.
>
> - *Was the SFT training data also curriculum-generated? Without a fair comparison where both SFT and RL use identical curriculum data, it's difficult to isolate whether the gains come from the curriculum design itself or from the RL optimization.*
>
> The SFT baseline in the main paper does not use curriculum-generated data.
> To better assess the benefit of RL, we additionally trained SFT using a curriculum data strategy.
> The results are provided in the table below.
>
>
> | Method | Curriculum Data | InfoSeek | EVQA |
> |--------|------------------|----------|------|
> | SFT    | ×                | 29.5     | 25.1 |
> | SFT    | √                | 30.5     | 34.4 |
> | DAPO   | ×                | 41.5     | 31.4 |
> | DAPO   | √                | 43.0     | 34.5 |
>
> We observe that while the curriculum data strategy improves SFT, RL still outperforms SFT under fair comparison, indicating that RL remains necessary.

---

> > ### Author Response · Authors · 2025-11-21
> >
> > ### W4: *Results are reported as single values without standard deviations or significance tests, making it difficult to assess the reliability of the comparisons.*
> >
> > We thank the reviewer for pointing this out.
> >
> > To address the concern, we have included a new section in the Appendix that assesses the reliability of our reported results. Specifically, we conducted each experiment **three times with independent runs** under the same training settings, recording performance on EVQA and InfoSeek at multiple training iterations.
> > Appendix Figure 5 (highlighted in blue) shows the performance curves for all three runs. We observe that while the **convergence speed** varies slightly across runs, the **final performance levels after convergence** are highly consistent.
> > These results indicate that our method is stable and the reported improvements are **robust to random initialization and training stochasticity**, addressing the reviewer’s concern about the reliability of our comparisons.
> >
> > ### W5: *Table 5's comparison may be misleading because it only lists training sample counts without reporting actual training time, GPU costs, or convergence speed. While Wiki-R1 achieves better results with less data, it employs computationally intensive RL training (multiple rollouts per sample, dynamic sampling, observation propagation), whereas baselines use simpler SFT. The paper reports Wiki-R1's training time but provides no training time data for baselines, making it hard to determine which method has lower total computational cost.*
> >
> > We thank the reviewer for the thoughtful feedback.
> >
> > Our key point is: Even after fully accounting for the RL overhead (e.g., multiple rollouts, dynamic sampling), Wiki-R1's total training cost is lower than most SFT baselines, making it efficient overall.
> > The updated table with training times confirms this:
> >
> > | Method      | FT Retrieval | FT Generation | #Training Sample (EVQA / InfoSeek) | Training Time (A100 hours) |
> > |-------------|--------------|---------------|-------------------------------------|-----------------------------|
> > | Wiki-LLaVA  | ×            | ✓             | 916,385 / 902,509                   | ~75                         |
> > | Echosight   | ✓            | ×             | 916,385 / 902,509                   | 40                          |
> > | ReflectiVA  | ×            | ✓             | 2,900,000 / 2,500,000               | ~1,688                      |
> > | Wiki-R1     | ×            | ✓             | 20,000 / 20,000                      | 36 (3B) / 48 (7B)           |
> >
> > Given that both Wiki-LLaVA and ReflectiVA are derived from the LLaVA-1.5 architecture and did not report their training times, we estimated their training costs based on LLaVA-1.5, using the formula:
> > Training Time ≈ Baseline Time × (Data Ratio / Batch Size Ratio) × Model Size Factor.
> > The results confirm that our method is not only data-efficient but also computationally more efficient overall.
> >
> >
> > ### Q1: *What exact data configuration was used for the SFT baseline in Table 3? Does it use the same entity-balanced samples as the RL experiments?*
> >
> > Please kindly refer to W3.

---

> ### Author Response · Authors · 2025-11-27
>
> Dear Reviewer kWfL,
>
> We truly appreciate your thoughtful feedback during the first review round. As the discussion period will close on December 3, we wanted to follow up and ensure that our clarifications have adequately addressed all the issues you raised. We would be glad to provide any further details if needed.
>
> If the concerns have been fully resolved, we would be grateful if you might consider adjusting the scores.
>
> Warm regards,
> Authors

---

### Author Response · Authors · 2025-12-03
**Overall Summary of Reviews and Rebuttal**

Dear Reviewers and Area Chair,

We sincerely thank the reviewers for their constructive feedback and are encouraged by the positive assessments of our work. The reviewers highlighted the following strengths:
- **Clear motivation and problem identification** — Our work clearly identifies the challenges RL faces in KB-VQA, particularly the impact of retrieval noise and sparse rewards (Reviewers kWfL, Wc3T, YGJ1).
- **Novel and elegant curriculum framework** — Reviewers praised the principled combination of data-level and sampling-level curricula, the controllable difficulty generation, and the use of Gaussian sampling with Observation Propagation to stabilize training (Reviewers kWfL, zx7D).
- **Strong empirical gains and notable data efficiency** — The method shows consistent improvements over baselines, strong generalization on unseen-question splits (e.g., +7.4 on InfoSeek Unseen-Q), and achieves these results with only ~40k samples, substantially fewer than prior work (Reviewers kWfL, Wc3T, zx7D).
- **Comprehensive ablations validating each component** — Ablation studies clearly demonstrate the necessity of both curriculum components and their contributions to improved RL training dynamics (Reviewer Wc3T).
- **Clear and polished presentation** — The reviewers noted that the paper is well-motivated and visually clear, including the main figure (Reviewer YGJ1).

Building on this positive feedback, we carefully addressed the reviewers’ concerns in our rebuttal and the revised paper:
- **Justification of partial control on retrieval noise.** Our partial control over retrieval noise has been shown to improve training stability. (Reviewer Wc3T W1)
- **Additional validation and robustness across settings.** We supplemented additional experiments to validate the effectiveness of our model on different base algorithms, retrieval settings, and datasets. (Reviewer Wc3T W3, Reviewer zx7D W2, Reviewer YGJ1 W5)
- **Clarification of the novelty of our methods.** The main novelty of our work is the introduction of an efficient difficulty-estimation mechanism for curriculum learning, which is performed in an unsupervised manner. (Reviewer zx7D W1)
- **Clarification of baseline settings.** We clarify that all baselines were trained under comparable conditions, ensuring a fair comparison. (Reviewer zx7D W4)
- **Additional discussion about relevant previous works.** We added discussions about more previous works and benchmarks in the revised paper. (Reviewer zx7D W3)
- **Specification of parameter settings and experiment configuration.** We clarify the hyperparameters used in our method and the detailed experiment configuration. (Reviewer YGJ1 W4)
- **Revision for clearer presentation of tables.** We revised the presentation of our tables to avoid potential confusion.(Reviewer YGJ1 W2)

Lastly, we note that a few points remain under active discussion with the reviewers. We value these suggestions as they help refine our framing, and we summarize the unresolved items and our ongoing clarifications below.
- **Additional experiments of hyperparameter transparency.** We conducted additional hyperparameter sensitivity experiments to validate the robustness of our method. In response to the reviewer’s second-round concerns regarding the experimental settings, we provide a detailed justification for the choices made. (Reviewer Wc3T W2)
- **Clarification of the effectiveness of the observation propagation component.** We first clarify the component’s effectiveness in terms of both efficiency and accuracy. We then provide a detailed description of the ablation experiment settings in Table 3 to help reviewers better understand the results. (Reviewer YGJ1 W3)
- **Explanation of our methods.** We first summarize our explanations of the methods. In the second-round reply, the reviewer specifically raised a concern regarding motivation. We clarify that the difficulty levels are designed to model realistic scenarios in which retrieval contains inherent noise.(Reviewer YGJ1 W1)

We also note that Reviewer kWfL did not take part in the subsequent discussion, and thus their points are not listed above. Nevertheless, we provided detailed responses to all of their comments in the rebuttal.

We hope this summary assists the Area Chair in quickly grasping the status of the discussions, and we appreciate the time and consideration devoted to our submission.

---

### Meta-Review · Area_Chair_oegp · 2026-01-07

**Summary:**

The authors have been highly responsive during the rebuttal, addressing most reviewer concerns by adding key experiments (generalization to ViQuAE and another RL algorithm, hyperparameter sensitivity) and clarifying methodological details and baseline fairness. While one reviewer (YGJ1) raised significant concerns, their critique regarding RAG's assumptions appears based on a misunderstanding, and their clarity concerns were not echoed by other reviewers. The primary remaining reservation pertains to the scope of the hyperparameter sensitivity analysis, which the authors have credibly justified.

Overall, the strengths—a clear problem, an elegant solution, and compelling results—outweigh the limitations. The paper offers a solid contribution that is likely to interest the ICLR community. A cautious acceptance is therefore recommended, contingent on the authors incorporating their promised minor revisions regarding the sensitivity analysis.

**Reviewer Concerns:**

--addressed--: The performance gains are mostly demonstrated on EVQA and InfoSeek. It’s unclear whether the framework generalizes to other KB-VQA settings (e.g., OK-VQA) or to different retrieval model architectures.

**Reviewer Scores:**

Regarding the comment that the paper lacks sufficient details to assess the true benefit of RL, the authors provided additional experimental results during the rebuttal that explicitly address this concern. Based on these clarifications, the AC believes that the reviewer would likely have revised their score had they been able to participate fully in the discussion.

---

### Decision · Program_Chairs · 2026-01-26

Accept (Poster)